# Tradeoff between speed and robustness in primordium initiation mediated by auxin-CUC1 interaction

Shuyao Kong [1,2], Mingyuan Zhu[1,2,4,5], David Pan[1,2], Brendan Lane [3], Richard S. Smith [3] & Adrienne H. K. Roeder [1,2] ✉

Robustness is the reproducible development of a phenotype despite stochastic noise. It often involves tradeoffs with other performance metrics, but the mechanisms underlying such tradeoffs were largely unknown. An *Arabidopsis* flower robustly develops four sepals from four precisely positioned auxin maxima. The *development related myb-like 1* (*drmy1*) mutant generates noise in auxin signaling that disrupts robustness in sepal initiation. Here, we find that increased expression of *CUP-SHAPED COTYLEDON1* (*CUC1*), a boundary specification transcription factor, in *drmy1* underlies this loss of robustness. CUC1 surrounds and amplifies stochastic auxin noise in *drmy1* to form variably positioned auxin maxima and sepal primordia. Removing CUC1 from *drmy1* provides time for noisy auxin signaling to resolve into four precisely positioned auxin maxima, restoring robust sepal initiation. However, removing CUC1 decreases the intensity of auxin maxima and slows down sepal initiation. Thus, CUC1 increases morphogenesis speed but impairs robustness against auxin noise. Further, using a computational model, we find that the observed phenotype can be explained by the effect of CUC1 in repolarizing PIN FORMED1 (PIN1), a polar auxin transporter. Lastly, our model predicts that reducing global growth rate improves developmental robustness, which we validate experimentally. Thus, our study illustrates a tradeoff between speed and robustness during development.

Cells within an organism experience a multitude of noise, such as stochastic gene expression[1] and heterogenous growth rate[2,3]. Despite such noise, organisms often develop invariantly and reproducibly, a phenomenon termed developmental robustness[4]. How developmental robustness is achieved is one of the most intriguing open questions in cell biology[5], which has attracted increasing research efforts in recent years[2,3,6–18]. It was shown that noise in morphogen signaling can be buffered by certain gene regulatory network structures[13,14] and self-organized cell sorting[11,15] to achieve robust patterning. Noise in growth rate can be buffered by averaging growth among neighboring cells[2,3] or in the same cell over time[2] to achieve robust organ size and shape. Notably, robustness is often involved in tradeoffs with other important aspects of development[19–21]. For example, during porcine embryogenesis, manual removal of zona pellucida speeds up development by a few hours but significantly reduces the robustness of blastocyst development in terms of symmetry and cell size uniformity[21]. How such tradeoffs are mediated remains largely unknown.

[1]Weill Institute for Cell and Molecular Biology, Cornell University, Ithaca, NY 14853, USA. [2]Section of Plant Biology, School of Integrative Plant Science, Cornell University, Ithaca, NY 14853, USA. [3]Department of Computational and Systems Biology, John Innes Centre, Norwich NR4 7UH, UK. [4]Present address: Department of Biology, Duke University, Durham, NC 27708, USA. [5]Present address: Howard Hughes Medical Institute, Duke University, Durham, NC 27708, USA. ✉e-mail: ahr75@cornell.edu

In plants, developmental robustness has been studied in sepals[2,3,16–18,22]. Sepals are the outermost floral organs that enclose and protect the immature bud before the flower opens. To achieve this protection, each flower robustly develops four sepals of constant size, positioned evenly around the bud typical of a cruciferous flower. This robustness in size, number, and position ensures tight closure critical for protection (Fig. 1a). In contrast, the *development related myb-like 1* (*drmy1*) mutant flower produces 3–5 sepals of different sizes, unevenly positioned, leaving gaps that expose the inner floral organs (Fig. 1a)[16,17]. This loss of developmental robustness originates during the initiation of sepals from the floral meristem, where the sepal primordia are robust in size, number, and position in wild type (WT) but variable in *drmy1* (Fig. 1b)[16]. We have been studying how DRMY1 maintains robust sepal development. We previously showed that DRMY1 maintains robustness by increasing TARGET OF RAPAMYCIN (TOR) signaling and mRNA translation, which supports the rapid synthesis of A-type ARABIDOPSIS RESPONSE REGULATOR (ARR) and Arabidopsis thaliana HISTIDINE PHOSPHOTRANSFER PROTEIN6 (AHP6) proteins to dampen cytokinin signaling. A proper level of cytokinin signaling ensures robust patterning of auxin signaling and sepal initiation. In *drmy1*, the lack of A-type ARR and AHP6 proteins and the consequent upregulation of cytokinin signaling increase stochastic noise in auxin patterning, underlying variable sepal initiation[17]. However, whether there are any tradeoffs between robustness and other properties of development in this system remains unknown.

Here, we found that a tradeoff exists between robustness and speed of sepal initiation from the floral meristem. In WT, strong, robust auxin maxima restrict the expression of *CUP-SHAPED COTYLEDON1* (*CUC1*), encoding a boundary-specifying transcription factor[23,24], to precise boundary domains immediately outside the auxin maxima. CUC1 increases the intensity of auxin maxima it surrounds, and promotes rapid sepal initiation. In *drmy1*, lack of robustness in auxin patterning causes an expansion of *CUC1* expression. CUC1 amplifies stochastic auxin noise in the *drmy1* floral meristem, forming variably positioned auxin maxima and sepal primordia. Removing CUC1 slows down sepal initiation but provides robustness against noise. Thus, the feedback interactions between auxin and CUC1 promotes rapid organogenesis under low noise conditions, but disrupts robustness under high noise conditions. Our study thus illuminates the mechanism behind the tradeoff between robustness and speed during organ initiation.

## Results

### *CUC1* is upregulated in *drmy1* mutant

To gain insights into key mechanisms controlling developmental robustness, we previously performed RNA-seq on inflorescence tissue of *drmy1* vs. WT at the stage of sepal primordium initiation[17]. We found

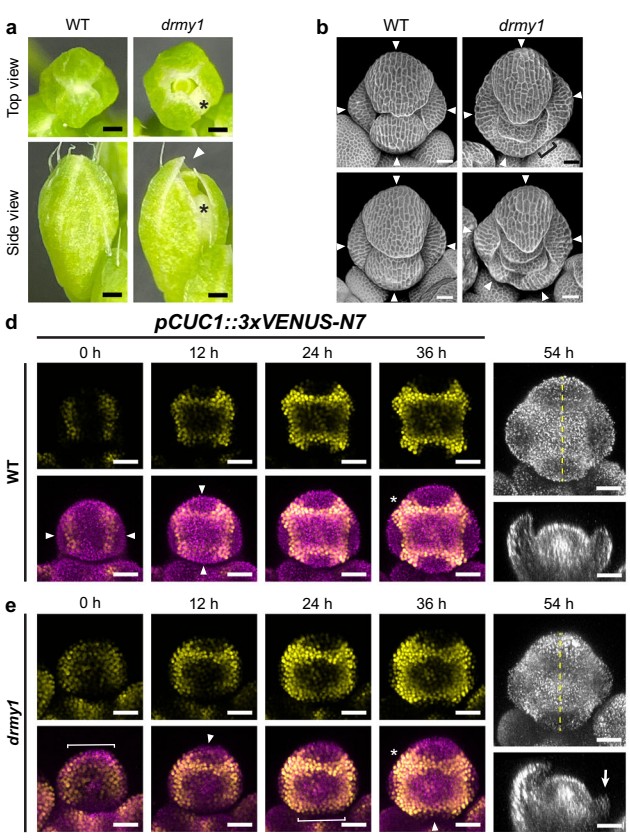

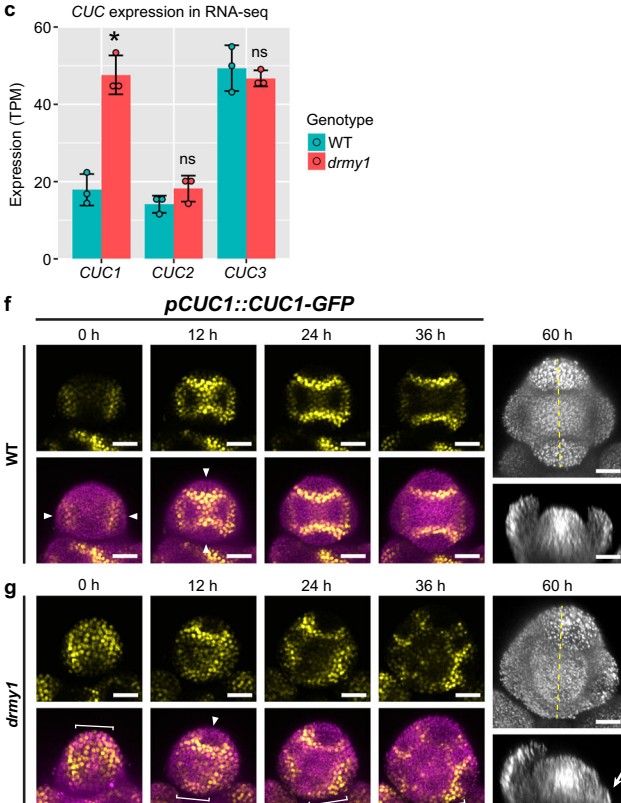

**Fig. 1 | CUC1 is upregulated in *drmy1* with disrupted spatial pattern. a, b** Floral phenotype of the *drmy1* mutant. **a** Stage 12 buds. Asterisk, a side gap caused by uneven sepal positions. Arrowhead, a top gap caused by unequal sepal length. **b** Stage 6 buds of WT (left), and *drmy1* buds with matching outer sepal size (right). Arrowheads, sepal primordia. Bracket, region without sepal outgrowth. **c** *CUC* expression in floral tissue of WT vs *drmy1* (in Ler *ap1 cal AP1-GR* background) (mean ± SD with individual data points). Transcripts per million (TPM). n = 3 per genotype. Adjusted p values from DESeq2: *CUC1*, 3.710 × 10⁻¹³; *CUC2*, 0.6650; *CUC3*, 0.2292. Source data are provided as a Source Data file. Complete dataset is in Kong et al.[17]. **d**–**g** Patterns of *CUC1* expression (**d, e**) and protein accumulation (**f, g**) in WT (**d, f**) vs. *drmy1* (**e, g**) flowers, live imaged. For 0–36 hours (h), top rows show the indicated CUC1 reporter; bottom rows show the CUC1 reporter (yellow) merged with Chlorophyll (magenta; to show flower morphology). For 54 or 60 h, the chlorophyll channel and its longitudinal section (dashed lines) are shown. Note that in WT, four boundaries with high CUC1 levels form at robust positions at stage 2 (arrowheads), corresponding to the robust initiation of four sepals at stage 3. In *drmy1*, *CUC1* expression expands to the bud periphery (brackets). This peripheral expression later narrows to boundary domains in some regions (arrowheads), and remains peripheral in other regions (brackets). This peripheral expression correlates with limited sepal outgrowth at stage 3 (arrows). *CUC1* expression in the intersepal regions are also expanded in *drmy1* compared to WT (asterisks). See Supplementary Movies 1 and 2. These image series are representative of n = 4 (**d**), n = 5 (**f**), n = 2 (**e**), and n = 4 (**g**) buds. Note that the *pCUC1::3xVENUS-N7* reporter is heterozygous in **d, e**. Scale bars in **a**, 250 μm; in other panels, 25 μm.

a 2.4-fold increase in the expression of *CUC1* in *drmy1*, but not for its paralogs *CUC2* and *CUC3* (Fig. 1c). *CUC* genes encode NAC (NAM, ATAF1,2, CUC2) family transcription factors important for boundary specification[23,24]. To investigate where *CUC1* is upregulated within the floral meristem, we imaged the *CUC1* transcriptional reporter. In WT, *CUC1* is expressed in four precisely specified boundaries separating the incipient sepal primordia from the center of floral meristem and each other, first appearing in the lateral boundaries followed by the outer and inner boundaries (Fig. 1d). In contrast, in *drmy1*, *CUC1* expression is expanded and localized to the bud periphery in early stage 2 meristems (Fig. 1e, Supplementary Movie 1). As the bud develops, this broadened expression coalesces into narrower, WT-like boundary domains in some parts of the bud (Fig. 1e, arrowheads), but remains in the bud periphery in other parts (Fig. 1e, brackets), correlated with the presence or absence of sepal outgrowth in stage 3. *drmy1* shows a similar disruption in the protein accumulation pattern of CUC1 (Fig. 1f, g, Supplementary Movie 2), as well as the expression and protein accumulation patterns of CUC2 (Supplementary Fig. 1).

It was previously shown that auxin inhibits the expression of *CUC* genes and restricts them to organ boundaries[25–27]. In the WT floral meristem, CUC1 accumulates in four boundaries immediately outside the four auxin maxima (Supplementary Fig. 2a). In *drmy1*, diffuse, bud periphery-localized CUC1 colocalizes with diffuse, weak bands of auxin signaling. As these auxin bands concentrate into variably positioned auxin maxima, CUC1 domains retreat from the bud periphery and refine into boundaries around the auxin maxima (Supplementary Fig. 2b). These observations led us to hypothesize that the broadened *CUC1* expression in *drmy1* is due to lack of robust, concentrated auxin maxima. Consistent with this idea, buds treated with L-Kynurenine (L-Kyn, inhibitor of auxin synthesis) or Naphthylphthalamic acid (NPA, inhibitor of polar auxin transport), both of which reduce auxin maxima (Supplementary Fig. 2c, d), show an expansion of *CUC1* expression into the bud periphery (Supplementary Fig. 2e, f). Treatment with both 1-Naphthaleneacetic acid (NAA) and NPA, which uniformly increases auxin signaling around the bud periphery (Supplementary Fig. 2d), largely represses *CUC1* expression, and only a weak ring of *CUC1* expression immediately inside the bud periphery remains (Supplementary Fig. 2f). These results support the idea that robust, concentrated auxin maxima are required for the precise boundary expression of *CUC1*, and that *drmy1* shows expanded *CUC1* expression due to diffuse auxin signaling.

## *CUC1* upregulation is necessary and sufficient for variable sepal initiation

We next tested the phenotypic consequence of *CUC1* upregulation, by imaging plants in which the repression of *CUC1* by miR164[28–30] has been removed. Specifically, we imaged *miR164* mutants (*eep1*[30], *mir164abc*[29]), and plants carrying *CUC1* expression constructs in which the miR164 target sequence has been mutated (*5mCUC1*[28] and *CUC1m-GFP*[30]). In WT, buds robustly develop four sepal primordia that are evenly spaced (Figs. 1b and 2a)[16,17]. We found that this robustness is disrupted in buds upregulating *CUC1*, which, similar to *drmy1*, produce a range of 2-6 sepal primordia that are unevenly spaced and of different sizes (Fig. 2b–e, Supplementary Fig. 3a–h). In WT, uniform sepal size within each flower is achieved by coordinated initiation timing between sepal primordia, where inner and lateral sepals initiate within 12 h of the outer sepal (Supplementary Fig. 4a, c)[16]. In contrast, in *drmy1*, variability of sepal size originates from the disorganized initiation timing, where the initiation of the inner and lateral sepals are severely delayed (Supplementary Fig. 4f, h, i)[16]. Similar to *drmy1*, we found that in buds upregulating *CUC1*, the initiation of inner and lateral sepals is greatly delayed relative to the outer sepal, underlying the variability in size (Supplementary Fig. 4a–c). In addition, the time difference between outer, inner, and lateral sepal initiation events is more variable between buds

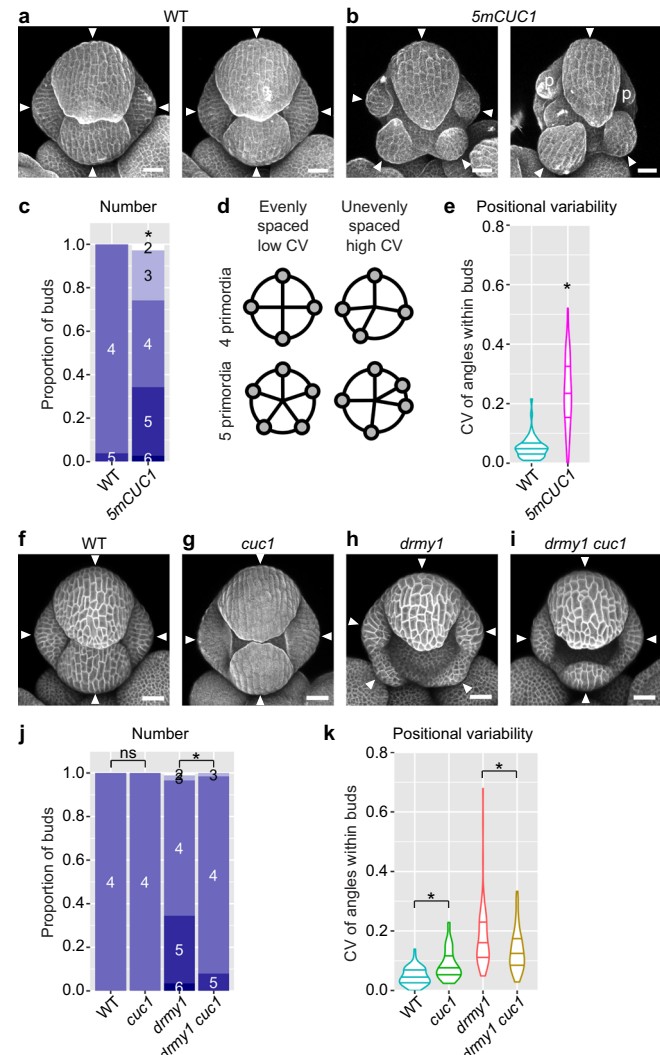

**Fig. 2 | *CUC1* upregulation is sufficient and necessary for variable sepal initiation. a–e** *CUC1* upregulation causes variable sepal initiation. **a, b** WT (**a**) and *5mCUC1* (**b**) buds, propidium iodide (PI)-stained. Arrowheads, sepal primordia. p, petal primordium. **c** Quantification of sepal primordium number in each bud. **d** How variability in sepal position was quantified. Buds in which sepal primordia are evenly distributed around the bud have low CV values of angular distances between adjacent sepal primordia (left), while buds in which sepal primordia are irregularly positioned have high CV values (right). Reproduced from Kong et al.[17] **e** Variability of sepal position in WT vs. *5mCUC1*. WT, n = 51 buds; *5mCUC1*, n = 70 buds. Asterisks indicate statistically significant differences in a Fisher's contingency table test (p = 4.535 × 10⁻¹⁴) (**c**) or Wilcoxon rank sum test (p = 2.878 × 10⁻¹⁶) (**e**). **f–k** The *cuc1* mutation rescues variability in sepal number and position in *drmy1*. **f–i** Buds of WT (**f**), *cuc1* (**g**), *drmy1* (**h**), and *drmy1 cuc1* (**i**). The *cuc1* bud was PI-stained, and others carried the *35S::mCitrine-RCI2A* membrane marker. Arrowheads, sepal primordia. **j, k** Sepal primordium number (**j**) and positional variability (**k**) quantified as described above. WT, n = 66 buds; *cuc1*, n = 52 buds; *drmy1*, n = 87 buds; *drmy1 cuc1*, n = 63 buds. Asterisks indicate statistically significant differences in a Fisher's contingency table test (**j**) or Wilcoxon rank sum test (**k**), and ns means no significant differences. P values for **j**: WT vs. *cuc1*, p = 1; *drmy1* vs. *drmy1 cuc1*, p = 2.575 × 10⁻⁴. P values for **k**: WT vs. *cuc1*, p = 2.665 × 10⁻⁶; *drmy1* vs. *drmy1 cuc1*, p = 4.922 × 10⁻³. Scale bars, 25 μm. Source data are provided as a Source Data file.

(Supplementary Fig. 4c). Overall, these results suggest that *CUC1* overexpression is sufficient to disrupt the robustness in number, position, and coordinated initiation timing of sepal primordia.

Our results show that *CUC1* overexpression is sufficient for disrupting robustness in sepal initiation, but is it also necessary? We found that the *drmy1 cuc1* double mutant often robustly develops four

sepal primordia that are evenly spaced, rescuing the variability in sepal number and position in *drmy1* (Fig. 2f–k). The result is specific to *cuc1* since the *drmy1 cuc2* double mutant exhibits disrupted robustness similar to *drmy1* (Supplementary Fig. 3i–n). While the *cuc1* mutation restores robustness in sepal primordia number and position in *drmy1*, it does not restore robustness in sepal primordia size and coordination of initiation timing (Supplementary Fig. 4d–i), suggesting that other mechanisms also contribute to these defects. Overall, these results show that *CUC1* upregulation causes the variability of sepal primordium number and position in *drmy1*.

## CUC1 increases the intensity of auxin maxima and facilitates rapid sepal initiation

CUC1 functions partially redundantly with CUC2 and CUC3 in organ separation[31,32], and the *cuc1 cuc2* double mutant shows sepal fusion in stage 8 flowers[31]. However, *cuc1 cuc2* flowers robustly initiate four sepal primordia at stage 4, suggesting that they are dispensable for sepal initiation. Then why is *CUC1* still expressed so early on in the floral meristem, when it has the potential to disrupt robustness when dysregulated (Fig. 2)? Comparing buds of similar size (as an indicator of similar developmental progression), we found that overexpression of *CUC1* makes the outer sepal primordium initiate earlier from the floral meristem (Fig. 3a–c). This increased speed in outer sepal initiation correlates with an increase in auxin signaling (Fig. 4a, b). Mutation of *CUC1* delays the initiation of all four sepals relative to bud size (Fig. 3d–f), correlated with weaker auxin signaling maxima (Fig. 4a, b). These results suggest that CUC1 increases the intensity of auxin maxima, which in turn promotes rapid sepal initiation to promptly cover and protect the developing floral meristem. This beneficial role may explain why *CUC1* is expressed in the early-stage floral meristem despite its potential in reducing developmental robustness when dysregulated, and suggests a potential conflict between speed and robustness in sepal initiation.

## CUC1 amplifies sporadic auxin noise into variably positioned auxin maxima

How does CUC1 mislocalization disrupt developmental robustness in *drmy1*? There is evidence that feedback interaction between CUC and auxin plays an important role in plant morphogenesis[25], and we previously showed that the disrupted pattern of auxin signaling underlies variable sepal initiation in *drmy1*[16]. Our results illustrated the first half of the feedback loop where auxin regulates CUC1, by showing that diffuse auxin signaling underlies the mislocalization of CUC1 expression in *drmy1* (Supplementary Fig. 2). Here, we hypothesized the second half of the feedback loop where CUC1 regulates auxin, and tested whether CUC1 mislocalization disrupts robustness in auxin pattern in *drmy1*. In WT, four auxin maxima form robustly, marking four incipient sepal primordia (Fig. 4a, Outer, Inner, Lateral, Lateral). We found that this robustness is disrupted in *SmCUC1* (CUC1 overexpression), as additional auxin maxima form (Fig. 4a, arrow). The *cuc1* single mutant forms four auxin maxima robustly, although weaker (Fig. 4a, b). The *drmy1* single mutant, where CUC1 is upregulated and mislocalized, shows diffuse, noisy bands of auxin signaling (Fig. 4a, brackets). Removing CUC1 from *drmy1* restores four robustly positioned auxin maxima (Fig. 4a, c, d). Overall, these results are consistent with the second half of the feedback loop where CUC1 mislocalization disrupts robustness in auxin pattern.

How is robust auxin pattern disrupted when *CUC1* is upregulated, and restored when *CUC1* is removed? To address this question, we live imaged buds of WT, *SmCUC1*, *cuc1*, *drmy1*, and *drmy1 cuc1* every 6 h from late stage 1 to early stage 3 (Fig. 4e–n). Initially, WT buds show three robustly positioned auxin maxima. One of them is in the cryptic bract (a suppressed inflorescence leaf), a remnant of floral meristem initiation[33–36]. The other two auxin maxima appear in the incipient lateral sepals. These auxin maxima are later followed by two more in

the incipient outer and inner sepals (Fig. 4e, j, Supplementary Movie 3). In *SmCUC1*, some buds initially form WT-like pattern of four auxin maxima, followed by additional ones in between (Fig. 4f, k, arrows, Supplementary Movie 4). Other buds display more dynamic spatio-temporal changes in auxin maxima localization (Supplementary Fig. 5). This suggests that *CUC1* overexpression disrupts auxin patterning by inducing the formation of new auxin maxima and making them more dynamic. In *cuc1*, auxin maxima sequentially and robustly form like in WT, although weaker (Fig. 4g, l, Supplementary Movie 5). In *drmy1*, auxin signaling initially appears as diffuse, noisy bands with sporadic patches of cells having stronger signal than neighboring cells which fluctuates over time (Fig. 4h, m, brackets). These sporadic patches seed the subsequent formation of variably sized and positioned auxin maxima (Fig. 4h, m, arrowheads). As the bud further expands, additional auxin maxima form, similar to *SmCUC1* (Fig. 4h, m, arrows, Supplementary Movie 6). In *drmy1 cuc1*, auxin signaling initially accumulates in sporadic patches, but unlike *drmy1*, they fade away, allowing the subsequent formation of the robust auxin pattern (Fig. 4i, n, asterisk, Supplementary Movie 7). Our results suggest that increased *CUC1* expression disrupts robust auxin patterning in *drmy1* by amplifying sporadic auxin noise to form variably positioned auxin maxima.

We next tested whether CUC1 amplifies sporadic auxin noise from sources other than the *drmy1* mutation. It was previously shown that exogenous cytokinin alters patterns of polar auxin transport[37,38], causing sporadic patches of PIN convergence and auxin signaling[16]. We hypothesized that the cytokinin-induced sporadic patches of auxin signaling would not be amplified in the *cuc1* mutant, and over-amplified in *CUC1* overexpression (*SmCUC1*) buds. Indeed, in WT buds, treatment with the synthetic cytokinin 6-Benzylaminopurine (BAP) amplifies sporadic auxin patches to form variably positioned auxin maxima and sepal primordia (Fig. 5a, b, g–k). While BAP induces sporadic auxin noise in early stage 2 buds of *cuc1*, it quickly fades away, and most buds robustly form four sepal primordia (Fig. 5c, d, g–k). In contrast, *SmCUC1* buds treated with BAP form numerous auxin maxima that often connect into a ring, which grow into 3-8 primordia, more variable than the mock-treated (Fig. 5e–k). Overall, these data support the idea that, under conditions that increase noise in auxin patterning, such as *drmy1* or under exogenous cytokinin treatment, CUC1 amplifies the noise to form variably positioned auxin maxima, which in turn disrupts robustness in sepal initiation. In the absence of CUC1, these noisy patches are not amplified and quickly fade away, leaving four robust auxin maxima.

## CUC1 promotes PIN1 repolarization prior to sepal initiation

It was previously reported that *CUC* genes promote PIN1 polarity in the developing leaf margin and ovule primordium[25,39]. We postulated that an increase in PIN1 polarity could explain the amplification of auxin noise in *drmy1* (Fig. 4e–n) and BAP-treated WT (Fig. 5b), as well as increased the intensity of auxin maxima in WT (Fig. 4a, b). We began by testing whether CUC1 affects PIN1 polarity in the floral meristem. We first imaged the PIN1-GFP reporter in WT and the *cuc1* mutant. At mid-stage 2, PIN1-GFP is highly polarized towards incipient sepal primordia in WT buds (Supplementary Fig. 6a). In contrast, this polarity pattern is less clear in *cuc1* mutant buds of similar size (Supplementary Fig. 6b). Towards late-stage 2, PIN1-GFP becomes highly polarized towards incipient sepal primordia in both WT and *cuc1* mutant buds (Supplementary Fig. 6c, d). These results suggest that CUC1 does not affect the final polarity of PIN1 but instead increases its speed of repolarization, likely in response to auxin that starts to accumulate in the incipient sepal primordia around mid-stage 2[17].

To further test whether CUC1 increases the dynamics of PIN1, we ablated WT and *cuc1* mutant buds carrying the PIN1-GFP reporter. It was previously found that, in response to ablation, PIN1 in the inflorescence meristem reorients away from the wound site starting from 2 h and completes reorientation within 4 h[40]. We thus used the ablation experiment as a test of the speed of PIN1 repolarization in stage 2 floral

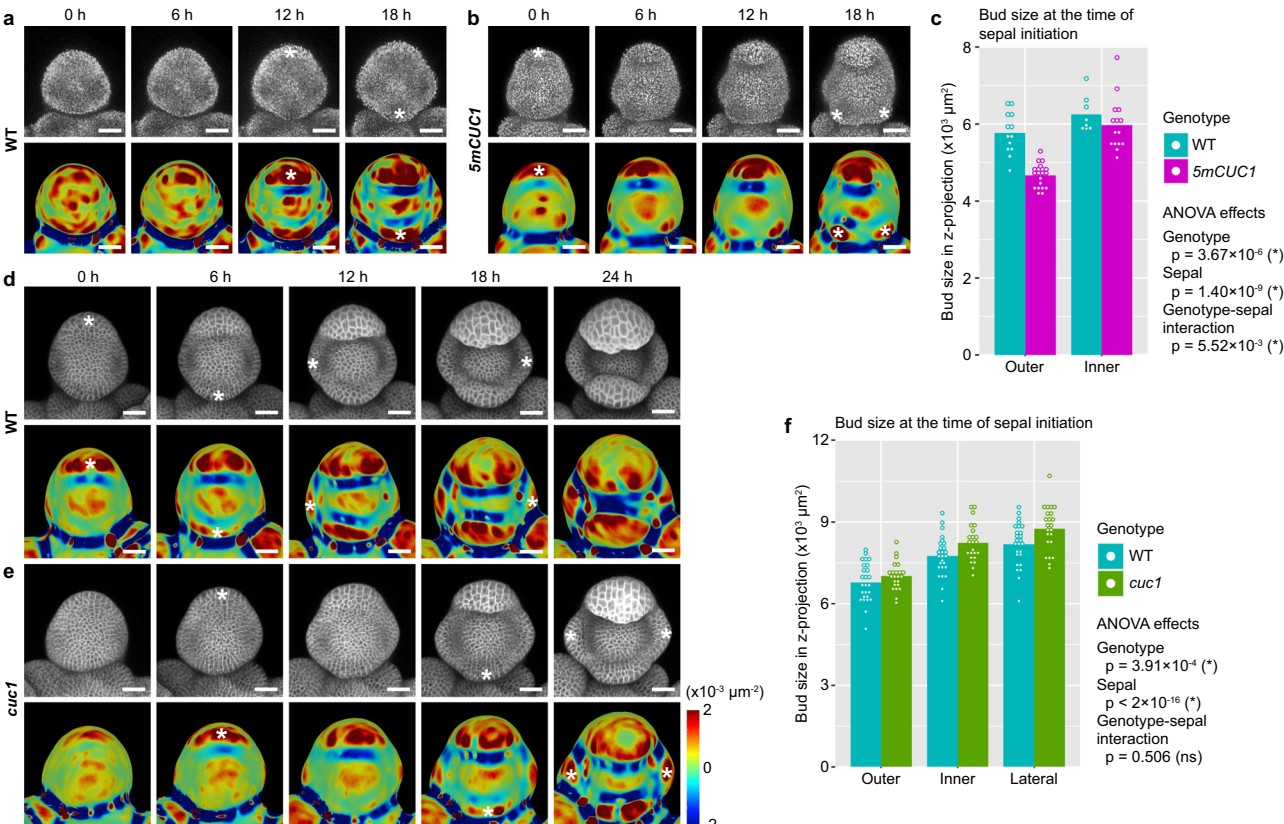

**Fig. 3 | CUC1 promotes rapid sepal initiation. a–c** The outer sepal initiates more rapidly in *5mCUC1* than WT. **a**, **b** A WT bud (**a**) and a *5mCUC1* bud (**b**) live imaged. Top rows, chlorophyll; bottom rows, Gaussian curvature of extracted surfaces. The buds were matched for size, indicating matched developmental stage. Asterisks, sepal initiation events. Note that outer sepal initiation was much earlier in *5mCUC1* than WT. **c** Quantification of bud size (area in Z-projections; as a proxy for developmental stage) at the time of sepal initiation. Shown are mean (bars) and individual buds (dots). Sample size, WT outer sepal, n = 13; *5mCUC1* outer sepal, n = 20; WT inner sepal, n = 8; *5mCUC1* inner sepal, n = 15. An ANOVA model of Size ~ Genotype * Sepal was fit. Genotype, degree of freedom (df) = 1, F = 26.823, p = 3.67 × 10⁻⁶. Sepal, df = 1, F = 53.946, p = 1.40 × 10⁻⁹. Interaction, df = 1, F = 8.387,

p = 5.52 × 10⁻³. **d–f** Sepals initiate more slowly in *cuc1* than WT. **d**, **e** A WT bud (**d**) and a *cuc1* mutant bud (**e**) live imaged. Top rows, the *35S::mCitrine-RCI2A* membrane marker; bottom rows, Gaussian curvature of extracted surfaces. The buds were matched for size, indicating matched developmental stage. Asterisks, sepal initiation events. Note that overall sepal initiation is delayed in *cuc1* compared to WT. **f** Quantification of bud size (area in Z-projections; as a proxy for developmental stage) at the time of sepal initiation. Shown are mean (bars) and individual buds (dots). Sample size, 26 buds for WT and 23 buds for *cuc1*. An ANOVA model of Size ~ Genotype * Sepal was fit. Genotype, df = 1, F = 13.198, p = 3.91 × 10⁻⁴. Sepal, df = 2, F = 60.760, p < 2 × 10⁻¹⁶. Interaction, df = 2, F = 0.685, p = 0.5055. Scale bars in all images, 25 μm. Source data are provided as a Source Data file.

meristems of WT and *cuc1*. In WT, half of the cells near the ablation site complete PIN1 reorientation within 1 h after ablation (Supplementary Fig. 6e). The average time for cells to complete reorientation was 1.49 ± 0.67 h (Supplementary Fig. 6g) and 1.45 ± 0.22 h when bud-wise averages were calculated (Supplementary Fig. 6h). This is faster than the reorientation speed previously reported in the inflorescence meristem[40]. In contrast to WT, cells in *cuc1* mutant buds show delayed PIN1 reorientation (Supplementary Fig. 6f), with an average reorientation time of 2.06 ± 0.80 h across cells (Supplementary Fig. 6g) and 2.08 ± 0.30 h for bud-wise averages (Supplementary Fig. 6h). Although slower than WT, at 4 h cells in *cuc1* mutant buds show a similar pattern of reorientation away from the wound site (Supplementary Fig. 6e, f). Overall, these results support the idea that CUC1 does not affect the final PIN1 polarity but makes it more dynamic in response to cues such as auxin or mechanical stress.

### Modeling predicts CUC1 amplifies auxin noise by repolarizing PIN1

We wondered whether an increase in PIN1 repolarization could explain the noise-amplifying effect of *CUC1*. To test this, we implemented a computational model of auxin pattern formation (Supplementary Data 1). The floral meristem was modeled using a 2D growing disk of cells, where cells divide when a size threshold is met. In each cell, PIN1

transports auxin to neighboring cells with high auxin concentrations. This up-the-gradient auxin transport has been shown to generate auxin maxima patterns from a homogeneous initial condition with small perturbations[41–43] and is a potential mechanism of auxin noise amplification. CUC1 is produced in cells with low auxin concentration. The sole function of CUC1 in the model is to increase PIN1 repolarization. Below a CUC1 concentration threshold, PIN1 repolarizes linearly according to the auxin concentration of neighboring cells; above the threshold, PIN1 repolarizes quadratically, and is thus more sensitive to auxin concentration differences among neighboring cells (Fig. 6a). We hypothesize that this function of CUC1 in increasing PIN1 repolarization can by itself promote auxin maxima formation while amplifying auxin noise. The simulation starts with a patternless disk of cells with small fluctuations in auxin production rate, which then forms auxin maxima that can be extracted and quantified (Fig. 6b). We modeled the *drmy1* mutation by increasing the amplitude of fluctuation in auxin production rate, which recreates the sporadic auxin patches and PIN1 convergence points observed in *drmy1*[16] (Fig. 6c). We modeled the *cuc1* mutant by eliminating CUC1 production (Fig. 6c). We found that in both WT and *drmy1* models, having CUC1 results in stronger, more concentrated auxin maxima that form more rapidly compared to *cuc1* and *drmy1 cuc1* respectively (Fig. 6d–h). These modeling results are similar to real buds (Fig. 4a, b). In both the model and the data, while

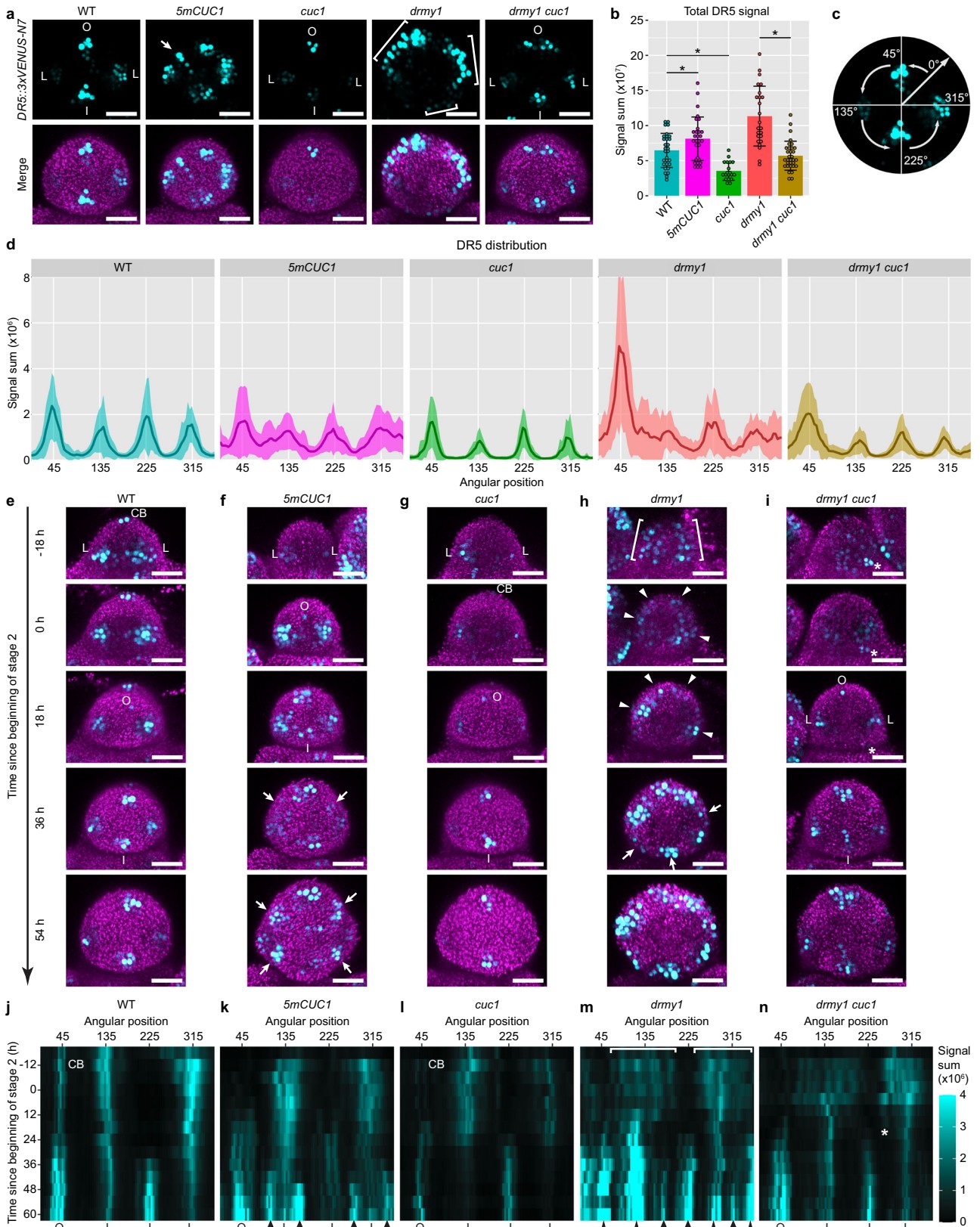

*drmy1* amplifies auxin production noise, creating variability in the final auxin pattern (Fig. 6f, arrowheads), *drmy1 cuc1* shows a relatively robust final pattern despite initial noise (Fig. 6g, arrows, 6i, Supplementary Movie 8). These simulations suggest that the role of CUC1 in increasing PIN1 repolarization (Supplementary Fig. 6) is sufficient for promoting rapid, strong auxin maxima formation while disrupting robustness against high auxin noise.

In addition to an increase in PIN1 repolarization which transports more auxin up the concentration gradient, noise may also be amplified if a cell rapidly grows and divides, forming daughter cells that inherit the same auxin noise, before noise is dampened by transport or decay (Fig. 7a). To test this idea, we examined the effects of tissue growth rate (which correlates with cell division rate in the model) on the robustness of auxin patterning. Comparing buds that develop to similar sizes

**Fig. 4 | CUC1 overexpression is necessary and sufficient for variability in auxin maxima patterning. a–d** Auxin patterning is disrupted in *SmCUC1* and *drmy1* but rescued in *drmy1 cuc1*. **a** *DR5::3xVENUS-N7* (cyan, top row) and DR5 merged with the chlorophyll channel (magenta, bottom row) in stage 2 buds of WT, *SmCUC1*, *cuc1*, *drmy1*, and *drmy1 cuc1*. **b** Total DR5 signal (mean ± SD with individual data points). Asterisks indicate statistically significant differences in two-tailed t-tests (WT vs. *SmCUC1*, p = 0.0266; WT vs. *cuc1*, p = 2.09 × 10⁻⁶; *drmy1* vs. *drmy1 cuc1*, p = 1.32 × 10⁻⁶). **c** Illustration of circular histogram analysis in **d** and **j–n**. Each bud was aligned so that the incipient outer sepal was on the top and the inflorescence meristem on the bottom. DR5 signal was quantified in 4°-bins around the Z-axis starting between the incipient lateral and outer sepal, so that the incipient outer sepal would be at 45°. **d** Circular histograms of DR5 (mean ± SD). WT, n = 30 buds; *SmCUC1*, n = 29 buds; *cuc1*, n = 19 buds; *drmy1*, n = 24 buds; *drmy1 cuc1*, n = 32 buds.

**e–n** *CUC1* upregulation promotes auxin maxima formation, but can amplify sporadic auxin patches. **e–i** Live imaging of stage 2 buds of WT, *SmCUC1*, *cuc1*, *drmy1*, and *drmy1 cuc1* carrying *DR5::3xVENUS-N7*. Shown is DR5 (cyan) merged with chlorophyll (magenta). On the left shows time relative to the beginning of stage 2 (second row), an indicator of developmental progression. **j–n** Kymographs showing DR5 signal through time, in the same buds as in (**e–i**). O, incipient outer sepal; I, incipient inner sepal; L, incipient lateral sepal; CB, cryptic bract. Brackets indicate diffuse bands of auxin signaling that later form distinct, variably positioned auxin maxima (arrowheads). Arrows indicate additional auxin maxima that form in the space between existing ones. Asterisks indicate a sporadic auxin patch that gradually disappears in *drmy1 cuc1*. Scale bars in all images, 25 μm. Source data are provided as a Source Data file.

under different growth rates in silico, increasing growth rate increases the variability of auxin maxima in both WT and *drmy1*, while reducing growth rate restores robust auxin patterning in *drmy1* (Fig. 7b, c, f, Supplementary Movie 9). In *drmy1* simulations under reduced growth rate, the initial stochastic noise in auxin concentration fades, allowing the formation of robust auxin pattern (Fig. 7c), similar to simulations of the *drmy1 cuc1* double mutant (Fig. 6g).

To test whether reducing growth rate can restore developmental robustness to *drmy1* buds in vivo, we cultured WT and *drmy1* buds on low sucrose media (0.1% w/v) compared to the normal media with 1% (w/v) sucrose. Low sucrose media reduced tissue growth rate by half (Supplementary Fig. 7a). Under either condition, WT buds form four auxin maxima that are robustly positioned (Fig. 7d, g), which correspond to the robust initiation of four sepal primordia at these positions (Fig. 7d, i, Supplementary Fig. 7b). Under the normal condition, *drmy1* mutant buds amplify auxin noise to form incorrectly positioned auxin maxima (Fig. 7e, h), similar to our previous observations (Fig. 4h, m) and simulations (Fig. 6f). These incorrectly positioned auxin maxima give rise to incorrectly positioned sepal primordia (Fig. 7e, i, Supplementary Fig. 7b). In contrast, under the low sucrose condition with reduced tissue growth rate, the *drmy1* mutant bud shows gradual dampening of initial auxin noise and the formation of robust auxin pattern (Fig. 7e, h) similar to our simulations (Fig. 7b, c). The robust auxin pattern gives rise to robust initiation of four sepal primordia in most cases (Fig. 7e, i, Supplementary Fig. 7b). Overall, these results support the model prediction that reducing tissue growth rate restores developmental robustness in *drmy1* mutant buds, and highlight our idea that speed and robustness can be conflicting sides of pattern formation during development.

It was previously shown that spatiotemporal averaging of cell heterogeneity can underlie tissue-wide developmental robustness[2,44,45]. We hypothesize that reduced PIN1 sensitivity to fluctuating auxin levels in neighboring cells (Fig. 6g) or reduced growth rate (Fig. 7b, c) restores robust auxin patterning because they allow more time for auxin noise to average to concentrations similar to nearby cells. We deduce that setting auxin noise temporally (but not spatially) unchanging would eliminate this averaging (Supplementary Fig. 8a), and thus mutating *cuc1* or reducing tissue growth rate would no longer rescue the *drmy1* patterning defect. Indeed, when noise is set temporally unchanging, both *drmy1* and *drmy1 cuc1* shows stabilization of initial auxin noise into variably positioned auxin maxima, as does *drmy1* under reduced growth rate (Supplementary Fig. 8b–d, Supplementary Movie 10). Overall, these results support our idea that increased expression of CUC1 disrupts robustness in auxin patterning by increasing the sensitivity of PIN1 to fluctuating auxin levels, which hinders temporal noise averaging and promotes noise amplification.

## Discussion

Developmental robustness has fascinated biologists for over 80 years[46], yet the underlying mechanisms have just begun to be explored[2–4,6–18]. Here, we elucidated a mechanism through which the developmental robustness of sepals is shaped by DRMY1-auxin-

CUC1 interaction (Fig. 8). In the floral meristem, DRMY1 maintains robust auxin patterning[16] which in turn restricts the expression of *CUC1* to precise boundary domains adjacent to auxin maxima (Fig. 1d, f, Supplementary Fig. 2). CUC1 increases the intensity of auxin maxima and promotes rapid sepal initiation (Figs. 3 and 4a, b). In the *drmy1* mutant, diffuse, noisy auxin signaling causes an upregulation and expansion of the *CUC1* expression domain (Fig. 1e, g, Supplementary Fig. 2). CUC1 amplifies sporadic auxin patches to form variably positioned auxin maxima and sepal primordia (Figs. 4 and 5). In *5mCUC1* where *CUC1* is overexpressed, robustness in auxin pattern is disrupted even without mutations or treatments that increase auxin noise (Figs. 4a and 5e, Supplementary Fig. 5). BAP treatment that increases auxin noise further enhances this loss of robustness (Fig. 5f). When *CUC1* is mutated, auxin noise induced by BAP or the *drmy1* mutation have time to average out, allowing robust auxin pattern formation (Figs. 4i, 5d and 6g, Supplementary Fig. 8) and robust sepal initiation (Fig. 2i).

In an Arabidopsis flower, sepals enclose and protect the inner, developing floral organs before the flower opens. To achieve this function, they need to not only rapidly initiate from the floral meristem to promptly cover it, but also develop robustly so as to not leave any gaps (Fig. 1a). We found that these two traits, speed and robustness, are conflicting sides of sepal development, and that this tradeoff is mediated by auxin-CUC1 interaction (Fig. 8). CUC1 promotes strong auxin maxima formation and rapid sepal initiation, but also stabilizes auxin noise and can therefore disrupt robustness. On the other hand, lack of CUC1 slows down sepal development, but also allows time for noisy auxin signaling to robustly converge. While we showed that the *cuc2* mutation did not restore developmental robustness to *drmy1* (Supplemental Fig. 3i–n), and previously published micrographs showed that *CUC3* was expressed at a very low level in the floral meristem-incipient sepal boundary[47], we cannot exclude the possibility that CUC2 and CUC3 are also involved in this tradeoff alongside CUC1.

Our computational modeling suggests that the speed-robustness tradeoff can be fully explained by the function of CUCs in increasing PIN1 repolarization, which was previously reported in other developmental contexts[25,39] and tested here in the floral meristem (Supplementary Fig. 6). How CUCs increase PIN1 repolarization remains unknown. It was shown that polarity of PIN proteins can be regulated by phosphorylation (e.g., PID[48], D6PK[49], and PP2A[50]) or membrane trafficking (e.g., ABCB19[51] and ROP2[52]). Thus, CUCs may increase PIN1 repolarization by changing the expression of these important PIN regulators. Alternatively, CUCs may also inhibit growth, causing mechanical conflict with adjacent fast-growing regions which alters PIN1 polarity[40]. Further study is needed to test whether CUCs increase PIN1 repolarization by any of these mechanisms.

Is morphogenesis speed always involved in a tradeoff with robustness? Can a system achieve both aspects simultaneously? Earlier theoretical studies on the mammalian olfactory epithelium suggest that tissue regeneration regulated by a feedback loop cannot achieve

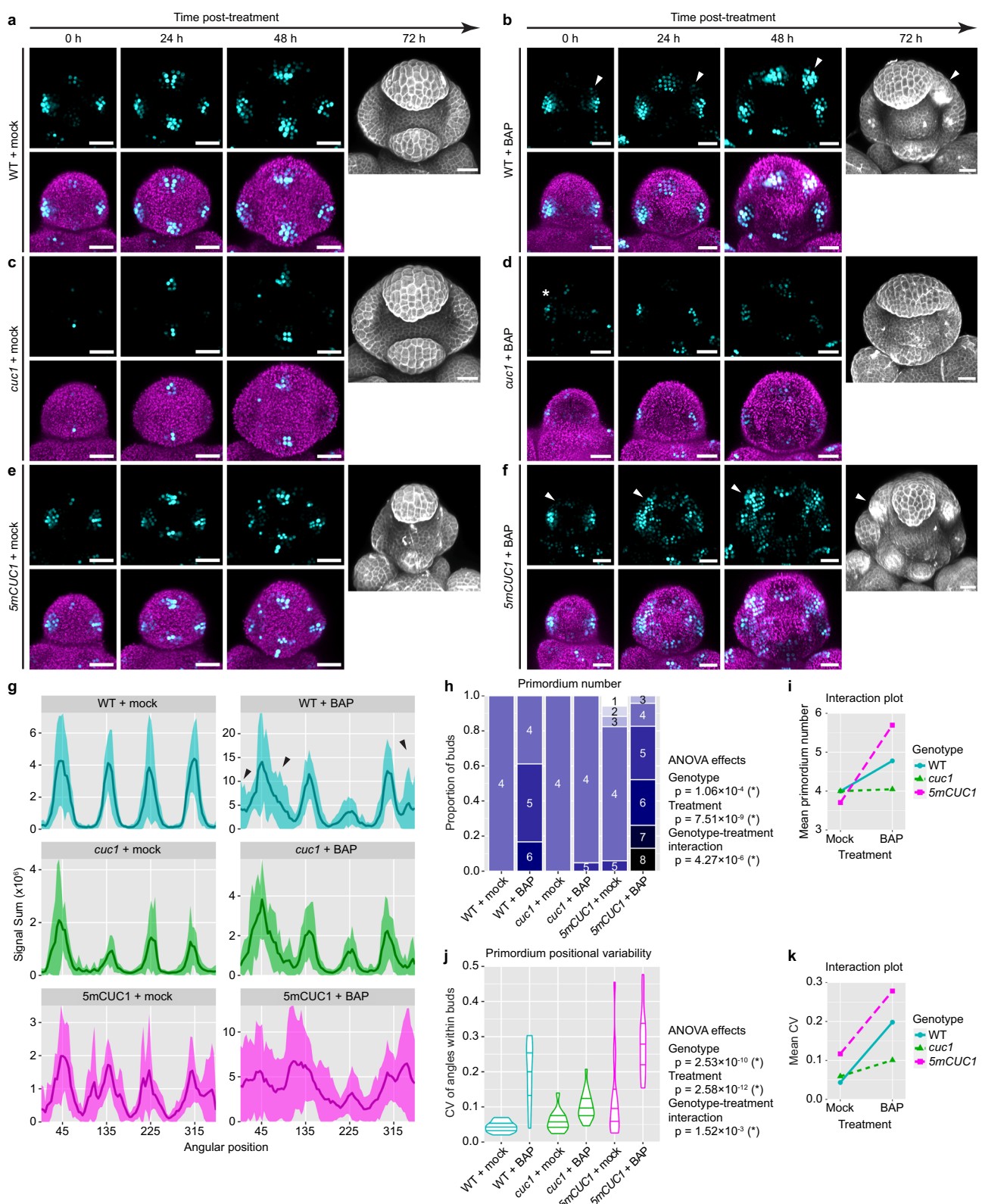

both speed and robustness, unless a second feedback loop is involved[19,20]. Similarly, a recent study on Hedgehog signaling shows that coupled extracellular and intracellular feedback loops mediated by Patched (PTCH) provide both robustness and speed of signaling gradient formation, compared to single or uncoupled feedback loops[13]. In summary, at the cost of additional regulatory mechanisms, morphogenesis speed and robustness may be both achieved simultaneously.

## Methods

### Plant material

Most Arabidopsis plants were in Col-0 background (WT). Plants presented in Supplementary Figs. 1 and 3a–c, e, f were in Ler background. *mir164abc* was in a mixed Ler-Col background[29]. *drmy1* (Ler) was generated by backcrossing *drmy1* (Col-0) with Ler twice. The following lines were provided by the Arabidopsis Biological Resource Center: *pCUC2::3xVENUS-N7* (CS23891)[53],

**Fig. 5 | CUC1 amplifies sporadic auxin patches to form variably positioned auxin maxima and primordia. a–f** WT (**a**, **b**), *cuc1* (**c**, **d**), and *5mCUC1* (**e**, **f**) were treated with mock (DMSO) (**a**, **c**, **e**) or 1 μM BAP (**b**, **d**, **f**) for 32 h and then transferred onto non-treatment media (0 h) and live-imaged every 24 h for four time points. For 0–48 h, top rows show *DR5::3xVENUS-N7* (cyan) and bottom rows show DR5 merged with Chlorophyll (magenta). For 72 h, buds were PI-stained. Note that BAP treatment disrupts robust auxin patterning and causes sporadic auxin patches. In *cuc1*, sporadic patches disappear (asterisk), whereas in WT and *5mCUC1*, they are amplified to form variably positioned auxin maxima and sepals (arrowheads). Scale bars, 25 μm. **g** Circular histograms of DR5 at 48 h post-treatment (mean ± SD). Note that BAP treatment causes regions of great variability in WT (large SD, black arrows), which is less pronounced in *cuc1* and more pronounced in *5mCUC1*. WT mock, n = 11 buds. WT BAP, n = 17 buds. *cuc1* mock, n = 11 buds. *cuc1* BAP, n = 19

buds. *5mCUC1* mock, n = 13 buds. *5mCUC1* BAP, n = 20 buds. **h–k** Quantification of sepal initiation pattern in stage 3 and 4 buds at 72 h post-treatment. **h** Primordium number in each bud. ANOVA model, Number ~ Genotype * Treatment. Genotype, df = 2, F = 10.04, p = 1.06 × 10⁻⁴. Treatment, df = 1, F = 39.81, p = 7.51 × 10⁻⁹. Interaction, df = 2, F = 14.01, p = 4.27 × 10⁻⁶. **j** Variability in primordium position. ANOVA model, CV ~ Genotype * Treatment. Genotype, df = 2, F = 27.790, p = 2.53 × 10⁻¹⁰. Treatment, df = 1, F = 63.585, p = 2.58 × 10⁻¹². Interaction, df = 2, F = 6.932, p = 1.52 × 10⁻³. Note that both sepal and petal primordia were counted, as they were hard to distinguish in many cases. **i**, **k** Interactions plots corresponding to (**h**, **j**). Note that *5mCUC1* further increases the variability caused by BAP treatment, while *cuc1* is largely resistant to the effect of BAP. WT mock, n = 13 buds. WT BAP, n = 18 buds. *cuc1* mock, n = 15 buds. *cuc1* BAP, n = 21 buds. *5mCUC1* mock, n = 17 buds. *5mCUC1* BAP, n = 23 buds. Source data are provided as a Source Data file.

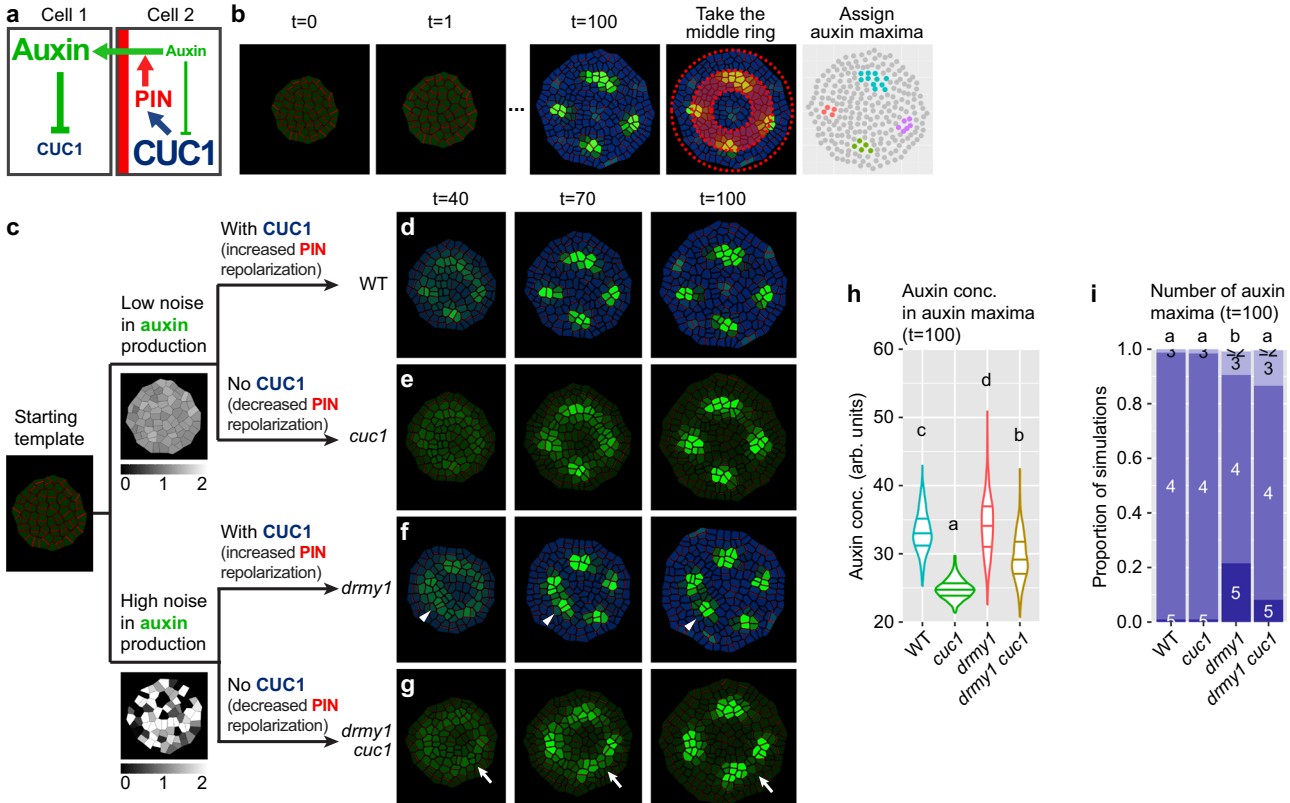

**Fig. 6 | The noise-amplifying effect of CUC1 can be fully explained by an increase in PIN repolarization. a** Diagram of auxin-CUC1 interaction implemented in the model. Auxin represses *CUC1* expression, whereas CUC1 increases PIN1 repolarization. PIN1 polarizes toward neighboring cells with high auxin concentrations and transports auxin into its neighbors. **b** Modeling approach. The floral meristem is modeled with a 2D disk of cells, which starts patternless. Molecular interactions in **a** generate auxin distribution patterns. Auxin maxima within the shaded middle ring (1/3 < r < 2/3 of total radius) were extracted and quantified. Green, auxin. Blue, CUC1. Red lines between cells, PIN1. **c** How genotypes were modeled. The *drmy1* mutation was modeled as having higher noise in auxin

production. The *cuc1* mutation was modeled by removing CUC1 (which decreased PIN1 repolarization). **d–i** Removing CUC1 rescues the variability in auxin patterning of *drmy1*. Shown are simulations of WT (**d**), *cuc1* (**e**), *drmy1* (**f**), and *drmy1 cuc1* (**g**). Note that while a sporadic auxin patch in *drmy1* was amplified to form an auxin maximum (arrowhead), a similar patch in *drmy1 cuc1* dissipated (arrow). Growth rate, 0.8. **h** Simulated auxin concentration averaged across all auxin maxima within each bud (violin plot with quartiles). **i** Number of auxin maxima in each bud at t = 100. n = 500 simulations per genotype. Letters show multiple comparison using Tukey's HSD. Source data are provided as a Source Data file.

*pCUC2::CUC2-VENUS pPIN1::PIN1-GFP* (CS67929)[53], *pCUC1::CUC1m-GFP* (CS65830)[30], *cuc1-13* (SALK_006496C), *cuc2-3* (CS875298), *eep1* (CS65826)[30], and *mir164a-4 mir164b-1 mir164c-1* (CS65828)[29]. In addition, the following lines were previously described: *drmy1-2*[16], *5mCUC1*[28], *pCUC1::3xVENUS-N7*[29], *pCUC1::CUC1-GFP*[54], *DR5rev::3xVE-NUS-N7*[53], *DR5rev::ER-mRFP1.2*[55], *35S::mCitrine-RCI2A*[16], *pPIN1::PIN1-GFP*[56]. *pCUC1::3xVENUS-N7* in Col-0 background (Fig. 1d, e) was generated by transforming Col-0 plants with the *pCUC1::3xVENUS-N7* construct.

## Plant growth conditions

Seeds were sown in wetted Lambert LM-111 soil and stratified at 4 °C for 3-5 days. Plants were grown under 16 h–8 h light–dark cycles (fluorescent light, 100 μmol m⁻¹ s⁻¹) at 22 °C in a Percival walk-in growth chamber.

## Flower staging

Flower buds were staged as previously described[57].

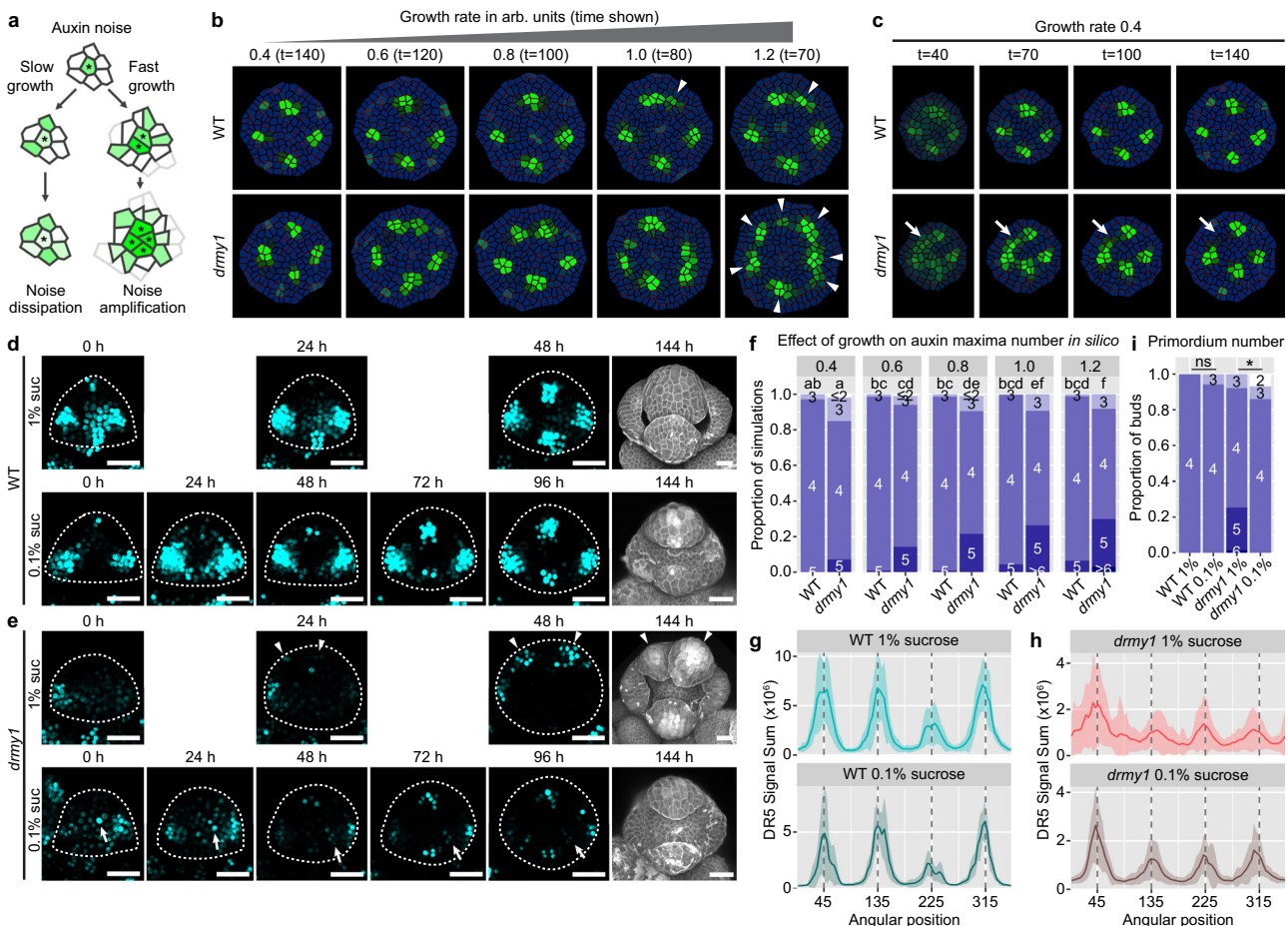

**Fig. 7 | Reducing growth rate restores developmental robustness in *drmy1*.**
**a** Reducing growth rate allows noise dissipation; while increasing growth rate promotes cell division, producing daughter cells inheriting the same auxin noise, amplifying the noise. **b** WT and *drmy1* simulations with different tissue growth rates from low (0.4) to high (1.2). Time points were chosen so that the resulting buds were of similar size. Note that slow growth restores robust auxin patterning in *drmy1*, while fast growth increases variability in both *drmy1* and WT, creating additional auxin maxima (arrowheads). **c** Time series of WT and *drmy1* simulations with low growth rate (0.4). Note that in *drmy1*, slow growth allows an initially diffuse band to canalize into robustly positioned auxin maxima (arrow). **d**, **e** Live imaging series of WT (**d**) and *drmy1* (**e**) buds cultured on 1% and 0.1% sucrose media. *DR5::3xVENUS-N7* signal is shown in cyan; bud contours are shown as dotted lines.

Arrowheads show amplification of auxin noise into incorrectly positioned auxin maxima and sepals. Arrows show noise dampening. Scale bars, 25 μm.
**f** Quantification of auxin maxima number in silico at time points corresponding to **b** across n = 500 simulations, comparing different growth rates. Letters show multiple comparison of mean auxin maxima number using Tukey's HSD.
**g**, **h** Circular histogram quantification of DR5 signal in vivo, in WT (**g**) and *drmy1* mutant (**h**) buds of similar size (~48 h for 1% sucrose and ~96 h for 0.1% sucrose) (mean ± SD). WT 1%, n = 10; WT 0.1%, n = 6; *drmy1* 1%, n = 30; *drmy1* 0.1%, n = 25.
**i** Number of sepal primordia initiated. WT 1%, n = 22; WT 0.1%, n = 17; *drmy1* 1%, n = 63; *drmy1* 0.1%, n = 43. Asterisks indicate statistically significant differences in Fisher's contingency table test (p = 1.677 × 10⁻⁴), and ns indicates no significant difference (p = 0.4359). Source data are provided as a Source Data file.

## Confocal microscopy

Confocal imaging was done as previously described[16,17]. Briefly, inflorescences were cut and dissected with a Dumont tweezer (Electron Microscopy Sciences, style 5, no. 72701-D) down to stage 9, inserted upright into a small petri dish (VWR, 60 × 15 mm) containing inflorescence culture medium (1/2 MS, 1% (w/v) sucrose, 1× Gamborg vitamin mixture, 0.1% (v/v) plant preservative mixture (Plant Cell Technology), 1% (w/v) agarose, pH 5.8), further dissected down to stage 6 (for static imaging) or stage 2 (for live imaging) and immersed with water. An exception was the low sucrose treatment, where the culture medium contained 0.1% (w/v) sucrose instead of 1%. The PIN1-GFP reporter was imaged using a Leica Stellaris 5 upright confocal microscope with a HC FLUOTAR L 25x/0.95 W VISIR lens, using counting acquisition mode and Lightning deconvolution. The rest of the images were acquired using a Zeiss710 upright confocal microscope with a ×20 Plan-Apochromat water-dipping lens (1.0 NA). For static imaging, to visualize tissue morphology, samples were stained with 0.1 mg/ml propidium iodide (PI) for 5 min before imaging. For live imaging, dissected samples were put in a 24 h-light growth chamber (fluorescent light, 100 μmol m⁻¹ s⁻¹) between

time points. To prevent bacterial growth, every 2-3 days, samples were transferred onto fresh media and treated with 100 μg/ml Carbenicillin (GoldBio, C-103-5, lot # 0129.091814 A).

The following lasers and wavelengths were used. Chlorophyll, excitation 488 or 514 nm, emission 660-722 nm (when also imaging mRFP1.2) or 647-721 nm (others). Propidium iodide, excitation 514 nm, emission 590-660 nm. mRFP1.2, excitation 561 nm, emission 582-657 nm. mCitrine, excitation 514 nm, emission 519-580 nm. VENUS in *DR5::3xVENUS-N7* and *pCUC2::CUC2-VENUS*, excitation 514 nm, emission 519-558 nm; in *pCUC1::3xVENUS-N7*, excitation 514 nm, emission 518-578 nm; in *pCUC2::3xVENUS-N7*, excitation 488 nm, emission 493-550 nm. GFP in PIN1-GFP, excitation 488 nm, emission 493-570 nm; in others, excitation 488 nm, emission 493-556 nm.

## Image processing

Image processing was done as previously described[16,17] and also briefly described below.

Tissue morphology was visualized by taking screenshots of the Chlorophyll, PI, or *35 S::mCitrine-RCI2A* channels in MorphoGraphX[58]

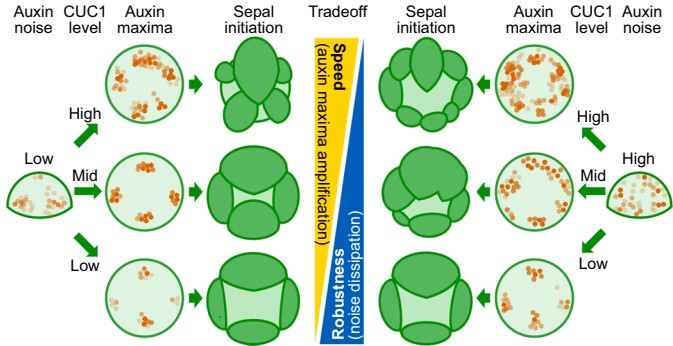

**Fig. 8 | A tradeoff exists between the speed and robustness of morphogenesis.** Strong expression of *CUC1* increases the intensity of auxin maxima and morphogenesis speed but disrupts developmental robustness by amplification of sporadic auxin noise. Mutating *CUC1* reduces morphogenesis speed but allows additional time for noise to dissipate and robust auxin pattern to form.

or by 3D rendering in the ZEN software (Processing → 3D). For Fig. 3 and Supplementary Fig. 4, to aid visualization of sepal initiation, Gaussian curvature heatmaps were calculated in MorphoGraphX as follows: Gaussian blur (X/Y/Z sigma = 1 μm twice and then X/Y/Z sigma = 2 μm once), edge detection (threshold = 2000-8000 depending on brightness, multiplier = 2.0, adapt factor = 0.3, fill value = 30000), marching cube surface (cube size = 8 μm, threshold = 20000), change lookup table to "jet", subdivide mesh, smooth mesh (passes = 5), subdivide mesh, smooth mesh (passes = 5), and project mesh curvature (type = Gaussian, neighborhood = 10 μm, autoscale = no, min curv = −0.002, max curv = 0.002). A sepal primordium is considered initiated when it emerges as a deep red band (positive curvature) separated from the floral meristem by a deep blue band (negative curvature).

For circular histograms and kymographs of DR5 signal, each stack was cropped in ImageJ and trimmed using the Voxel Edit function in MorphographX[58] so that only the focal bud remained in the stack. The bud was positioned so that it was centered, facing the Z direction, and the incipient outer sepal was at 45°. A circular histogram was calculated using the function Export Histogram Circular, summing signal (in voxel intensity units, 0-255) in each 1° sector around the Z axis starting from 0° (between the incipient outer and lateral sepals) counterclockwise. These histograms were then 4°-binned and used for plotting mean ± SD and kymographs. Total signal was calculated by summing all the bins.

### Quantification of developmental robustness
For variability in sepal primordium position, within each bud, an angle was measured between each pair of adjacent sepal primordia with respect to the bud center. CV was calculated within each bud as a measurement of how evenly sepal primordia are distributed around the bud.

For relative initiation timing of sepals within each bud, we considered that the robust temporal sepal initiation pattern in WT consists of the inner sepal initiating within 6 h of the outer sepal, and the lateral sepal initiating within 12 h of the outer sepal[16]. Severely delayed inner and lateral sepal initiation indicate loss of robustness. Thus, for each genotype, we calculated, among all buds, the mean and SD of initiation timing difference between inner and outer sepals, and between lateral and outer sepals. Robustness is considered lost if sepal initiation does not follow the WT temporal pattern (mean is greater than 6 and 12 respectively), and also if different buds have different temporal patterns (SD is large).

For sepal initiation timing relative to bud size, bud area (μm²) in maximum intensity projection images were measured in ImageJ[59,60] and used as a proxy for bud size. Representative

images of WT vs. *cuc1* (both carrying *35S::mCitrine-RCI2A*, a plasma membrane marker) that are of similar area and growth rate throughout the time series were shown. Similarly for WT vs. *5mCUC1*, although they did not have a membrane marker and thus the Chlorophyll channel was used for area measurement. Bud size at the time of outer, inner, and lateral (WT vs. *cuc1*) sepal initiation was plotted. For *5mCUC1*, not enough samples initiated lateral sepals, and thus bud size at the time of lateral sepal initiation was not analyzed. We reasoned that buds whose sepals initiate rapidly should be smaller when they initiate sepals, compared to buds whose sepals take longer to initiate. To control for the gradual reduction in size of successive buds during prolonged in vitro culture, only buds that did not have any sepals at the first time point and produced at least one sepal at or before the fourth time point (within 18 h of the first time point) were analyzed.

### In vitro drug treatments on inflorescence samples
For BAP treatment, inflorescences were dissected, put onto an inflorescence culture medium (see above) containing 0.05% DMSO and 1 μM BAP (6-Benzylaminopurine, Alfa Aesar, A14678). Mock contained just 0.05% DMSO. They were left in a growth chamber for 32 h and then transferred onto inflorescence culture medium without treatments, and imaged at 0 h, 24 h, 48 h, and 72 h after the transfer.

For L-Kyn treatment, inflorescences were dissected and put onto an inflorescence culture medium containing 0.02% DMSO and 80 μM L-Kyn (l-kynurenine, Sigma, K8625). Mock contained just 0.02% DMSO. They were left in a growth chamber for 4 days before imaging.

For NPA and NPA + NAA treatment, inflorescences were dissected and put onto inflorescence culture medium. The following solutions were made in water: for mock, 0.05% DMSO, 0.01% Silwet L-77; for NPA, 100 μM NPA (Naptalam, Sigma, 33371), 0.05% DMSO, 0.01% Silwet L-77; for NPA + NAA, 100 μM NPA, 20 μM NAA (1-Naphthaleneacetic acid, Sigma, N0640), 0.05% DMSO, 0.01% Silwet L-77. These solutions were applied on top of the dissected inflorescences for 24 h. Then, the solutions were discarded, and samples were washed three times with sterile water. Images were taken 48 h (for *DR5::3xVENUS-N7*) or 72 h (for *pCUC1::3xVENUS-N7*) after the end of the treatment.

### Ablation of floral meristem
Ablation was done using a pair of sharpened dissecting forceps (Electron Microscopy Sciences, style 5, no. 72701-D) on stage 2 floral meristems on dissected inflorescences. Floral meristems were imaged prior to ablation, ablated, immediately imaged after ablation, and imaged every hour for 4 h. In quantification of PIN1 reorientation timing, buds with less than ten cells responding to ablation were discarded; in the remaining buds, only cells that completed reorientation within 4 h were quantified.

### Mass-spring model of a growing floral meristem
We modeled the stage 2 floral meristem as a 2D disk of growing and dividing cells. The simulation begins as a round disk of 73 cells and 348 walls with no auxin, no CUC1, and apolar PIN. As detailed below, in each iteration, the following processes are run in order: deformation of cell walls under turgor pressure (repeated until convergence); dilution of auxin and CUC due to changes in cell size; updating noise in auxin production; 10 steps of chemical interactions; cell division; splitting of cell walls segments longer than 1 μm into smaller segments; cell growth; reinitialization; data output. The result of each iteration was used as the starting point of the next. Simulations were run for the desired number of iterations. Screenshots and data were saved every 10 iterations.

For tissue mechanics, a mass-spring model was used. Cell walls are subdivided to have a maximum length of 1 μm. Wall segments are

represented by springs in the simulation, with an initial resting length the same as in the starting configuration. Uniform turgor pressure is simulated by assigning a normal force to the boundary walls, as it cancels out on interior walls. The force acting on a vertex $\nu$ due to the springs, $\mathbf{F}_\nu$, was calculated as:

$$\mathbf{F}_\nu = \sum_{n \in N_\nu} k \left( \frac{||\mathbf{p}_n - \mathbf{p}_\nu||}{L_{\nu:n}} - 1 \right) \frac{\mathbf{p}_n - \mathbf{p}_\nu}{||\mathbf{p}_n - \mathbf{p}_\nu||} \tag{1}$$

where $N_\nu$ are neighboring vertices of vertex $\nu$, $\mathbf{p}_\nu$ is the position of vertex $\nu$, $\mathbf{p}_n$ is the position of a neighboring vertex $n$, $k$ is the spring constant (stiffness), and $L_{\nu:n}$ is the resting length of the spring joining $\nu$ and $n$[61,62]. Stiffness is set uniform and constant for all the cell walls. The resulting system of equations is solved using the backward Euler method with the GPU based stabilized bi-conjugate gradient solver available in MorphoDynamX. Growth was simulated by increasing the rest length of the springs based on their relative stretch, multiplied by an extensibility factor $g$:

$$\frac{dL_{\nu:n}}{dt} = g \left( \frac{||\mathbf{p}_n - \mathbf{p}_\nu|| - L_{\nu:n}}{L_{\nu:n}} \right) \tag{2}$$

Within each simulation, extensibility was set constant and uniform for all cell walls. Extensibility was changed when modeling meristems with increased or decreased growth rate.

The molecular interactions are that (a) auxin represses *CUC1* expression, and (b) CUC1 increases the sensitivity of PIN repolarization to auxin concentration differences among neighboring cells. To simulate them, each cell is assigned two fields, auxin concentration (aux) and CUC concentration (cuc). Each wall segment is assigned two fields, $pin_P$ and $pin_N$, denoting the amount of PIN protein on each side that mediates polar auxin transport in either direction. At each chemical step, three calculations take place concomitantly:

(a) For each cell $i$ at time step $t$, the amount of PIN on its wall facing neighboring cell $j$ is calculated as:

$$pin_{i \to j}(t) = (1 - \alpha) pin_{i \to j}(t-1) + \alpha \frac{aux_j^n L_j}{\sum_{k \in N_i} aux_k^n L_k} \tag{3}$$

$$pin_{i \to j}(0) = \frac{L_j}{\sum_{k \in N_i} L_k} \tag{4}$$

where $\alpha = 0.01$ is the PIN repolarization speed, $N_i$ is all the neighboring cells of cell $i$, $L_j$ and $L_k$ are lengths of cell walls of cell $i$ facing neighboring cells $j$ and $k$ respectively, and $aux_j$ and $aux_k$ are the auxin concentrations of cell $j$ and $k$ respectively. $n$ is the PIN sensitivity factor dependent on the CUC concentration of cell $i$, $cuc_i$, and the threshold CUC concentration, $cuc_{thres} = 2$:

$$n = \begin{cases} 1 \ (cuc_i < cuc_{thres}) \\ 2 \ (cuc_i > cuc_{thres}) \end{cases} \tag{5}$$

Note that the total amount of PIN in each cell sums to 1 over all its cell walls, unaffected by auxin concentration. Also note that a zero-flux boundary condition was used, i.e., no PIN is polarized towards the boundary of the modeled tissue, and there is no flux of auxin across the boundary.

(b) Change in auxin concentration in cell $i$ due to production, decay, and transport

$$\begin{aligned} \frac{d(aux_i)}{dt} &= Prod_{aux} \times \delta_{aux,i} - Dec_{aux} \times aux_i \\ &+ \frac{Tran_{aux} \sum_{j \in N_i} (aux_j \times pin_{j \to i} - aux_i \times pin_{i \to j})}{Area_i} \end{aligned} \tag{6}$$

where $Prod_{aux} = 1$ is the auxin production coefficient, $Dec_{aux} = 0.2$ is the auxin decay coefficient, $Tran_{aux} = 400$ is the polar auxin transport coefficient, $N_i$ are the neighboring cells of $i$, $Area_i$ is the area of cell $i$, $pin_{i \to j}$ is the amount of PIN on the wall of cell $i$ facing neighbor $j$, and $pin_{j \to i}$ is the amount of PIN on the wall of neighbor $j$ facing cell $i$. $\delta_{aux,i}$ is the auxin production noise of cell $i$, drawn during initialization and at each iteration from a Gaussian distribution $N(1, SD_{aux})$ where $SD_{aux} = 0.1$ for WT and *cuc1*, and $SD_{aux} = 1$ for *drmy1* and *drmy1 cuc1*, and negative values are set to 0. If the noise is set temporally unchanging (Supplementary Fig. 8), $\delta_{aux}$ values are drawn only during initialization but not at each iteration.

(c) Change in CUC concentration in cell $i$ due to production and decay

$$\frac{d(cuc_i)}{dt} = Prod_{cuc} \times \frac{1}{1 + \left( \frac{aux_i}{K_{aux}} \right)^{hill}} - Dec_{cuc} \times cuc_i \tag{7}$$

where $Prod_{cuc}$ is the CUC production coefficient (1 for WT and *drmy1*, and 0 for *cuc1* and *drmy1 cuc1*), $Dec_{cuc} = 0.2$ is the CUC decay coefficient, $K_{aux} = 5$ is the concentration of auxin at which CUC production is halved, and $hill = 4$ is an arbitrary Hill coefficient.

For cell division, a cell divides when its area passes a threshold of $50 \ \mu m^2$. Position of the new wall follows the minimal wall length principle, with noise parameters as follows (constant for all simulations): cell division noise 2.0, cell center noise 2.0, wall junction noise 2.0. Daughter cells inherit the same aux, cuc, and $\delta_{aux}$. Split cell walls inherit $pin_P$ and $pin_N$ proportional to their new lengths. New cell walls are assigned infinitesimal (non-zero) starting values of $pin_P$ and $pin_N$.

## Software

Image processing was done in ImageJ (version 2.14.0/1.54 f, build c89e8500e4)[59,60] and MorphoGraphX (version 2.0, revision 1-354, CUDA version 11.40)[58]. Modeling was implemented in C++ using vlab (version 5.0, build #3609)[63] and MorphoDynamX (version 2.0, revision 2-1395, CUDA version 11.40; www.MorphoDynamX.org) in an Ubuntu 20.04.6 LTS system, equipped with an Intel Core i9-10900 × 3.7 GHz 10-Core Processor, G.Skill Trident Z RGB 256 GB DDR4-3600 CL18 Memory, and a NVIDIA GeForce RTX 4090 24 GB Graphics Card. Data processing was done in RStudio (R version 4.3.1 (2023-06-16) -- "Beagle Scouts")[64]. Graphs were made using the package ggplot2 (version 3.4.2)[65]. Fisher's contingency table tests were done using fisher.test. Wilcoxon rank sum tests were done using wilcox.test. Levene's tests of homoscedasticity were done using leveneTest in package "car" (version 3.1-2). Data fitting with ANOVA was done using the function aov. Figures were assembled in Adobe Illustrator (version 27.8.1). An RGB color profile "Image P3" was used for all the figures.

## Statistical analysis

In most cases, each bud, either from the same inflorescence, a different inflorescence from the same plant, or a different plant, is considered a biological replicate. For RNA-seq presented in Fig. 1c, each RNA sample from 5 to 10 inflorescences of *ap1 cal AP1-GR* background (either WT or *drmy1*), extracted separately, is considered a biological replicate.

For bar plots of sepal primordium number or auxin maxima number, all buds (or simulations) were groups by genotype, and then grouped by number, and plotted as stacked bars. For bar-and-whisker

plots in Figs. 1c and 4b, bar shows mean, and whiskers show mean ± SD. For violin plots in Figs. 2e, k and 6h, Supplementary Fig. 3h, n, and 4c, h, i, horizontal lines show the quartiles. For circular histograms in Figs. 4d and 5g, lines show mean, and shaded area shows mean ± SD.

Student's *t* tests in Fig. 4b and Supplementary Fig. 6h, 7a and Wilcoxon's rank sum tests in Fig. 2e, k, Supplementary Fig. 3h, n, 4c, h, i and 7b are two-tailed, and compare only the indicated pairs of genotypes. For Levene's test of variability in Supplementary Fig. 4c, h, i, center of each group was calculated as mean. For ANOVA, formulae, degree of freedom, F, and p values are indicated in the Figure legends. For Tukey's HSD in Fig. 6h, i, 7f Supplementary Fig. 8d, all groups (combinations of genotype and growth rate) were fit using a linear model, Estimated Marginal Means (EMM) were calculated, and significant differences were indicated using compact letter display.

For RNA-seq, an FDR of 0.05 was used. For Tukey's HSD, family-wise type I error rate was 0.05. For all other analyses, a *p* value threshold of 0.05 was used.

### Reporting summary
Further information on research design is available in the Nature Portfolio Reporting Summary linked to this article.

## Data availability
The RNA seq data used in this study are available in the GEO database under accession code GSE230100. Source data are provided with this paper.

## Code availability
Source code for the computational model, a tutorial, and demo data are available in Supplementary Data 1 and on GitHub (https://github.com/RoederLab/MassSpringAuxin; https://doi.org/10.5281/zenodo.11526914)[66].

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

## Acknowledgements

We thank Isabella Burda, Trevor Cross, Michelle Heeney, Henrik Jönsson, Byron Rusnak, and Avilash Singh Yadav for their comments and suggestions on the manuscript. We thank Elliot Meyerowitz and Arnavaz Garda for the *DR5::3xVENUS-N7/PIN1::GFP* (Ler) seeds and the *pCUC1::3xVENUS-N7* construct. We thank Mitsuhiro Aida for the *pCUC1::CUC1-GFP* seeds. We thank Alexis Maizel for the *DR5::ER-mRFP1.2* seeds. We thank Teva Vernoux and Géraldine Brunoud for the *pPIN1::PIN1-GFP* seeds. We thank the Arabidopsis Biological Resource Center for providing seeds used in this research. We thank Addgene for providing the *5mCUC1 pGreenIIO129* construct (#12067). Research reported in this publication was supported by the National Institute of General Medical Sciences of the National Institutes of Health (NIH) under award number R01GM134037 to A.H.K.R., and a Biotechnological and Biological Sciences Research Council (BBSRC) Institute Strategic Program Grant (BB/X01102X/1) to R.S.S. The content is solely the responsibility of the authors and does not necessarily represent the official views of the National Institutes of Health.

## Author contributions

Conceptualization and design of experiments were done by S.K., M.Z., and A.H.K.R. Experiments were carried out by S.K. Data analysis was done by S.K. and D.P. The MorphoDynamX modeling platform was developed by B.L. and R.S.S. The floral meristem model was developed by S.K. based on scripts written by B.L. and R.S.S. Manuscript was prepared by S.K. and edited by all the authors. Funds were provided by A.H.K.R. and R.S.S.

## Competing interests

The authors declare no competing interests.
