## [Peer Review File · Nature Communications]

REVIEWER COMMENTS

Reviewer #1 (Remarks to the Author):

The manuscript entitled “Tradeoff Between Speed and Robustness in Primordium Initiation Mediated by Auxin-CUC1 Interaction” by Kong et al. describes the role of DRMY1-CUC1-Auxin in control of speed/robustness of floral organ formation by a precise manner. The authors show that CUC1 is upregulated in the drmy1 background. Removing of CUC1 may rescue the noise problem of the auxin signaling, while this may lead to a decreased speed of sepal initiation. The idea of the balance between speed and robustness in organ initiation seems to be very interesting. Overall, the data are very clearly presented and the manuscript is well organized.

Specific points:

- (1) The authors also tested the CUC2 expression pattern (Fig S1) and showed a similar result as CUC1. However, CUC2 could not function as CUC1 in genetic experiments. How about CUC3? Since the three CUC genes may play redundant roles in organ initiation, what is the unique molecular function for CUC1? Can CUC2 replace CUC1 in the cuc1 mutant background? The authors may have a deeper discussion on the CUC genes.
- (2) How about the upregulation of CUC2/CUC3? Could the upregulation of CUC2/CUC3 in regulation of organ initiation be similar to CUC1 upregulation?
- (3) I found that some of the data in Supplemental Figures are important for the story, for examples some data in Figure S6 and S7. Is it possible for some of them to be moved to the main text?

Reviewer #2 (Remarks to the Author):

The manuscript by Kong et al. entitled “Tradeoff Between Speed and Robustness in Primordium Initiation Mediated by Auxin-CUC1 Interaction” reports a role of the auxin-CUC module in patterning robustness. In the drmy1 mutant, in which sepal formation is less synchronized and can have variable number of sepals, the authors found that CUC1 expression was increased, and reducing CUC1 expression alleviates the phenotype. They also found that overexpressing CUC1 similarly reduced the robustness of sepal formation. Through live imaging, the authors showed that CUC1 expression promotes sepal primordium growth rate in a non-cell-autonomous manner, although CUC1 per se reduces cell growth. Auxin signaling maxima were also found enhanced, and ectopic auxin signaling, which forms proceed of sepal primordium formation, was observed in CUC1 overexpression lines and drmy1. The authors proposed a

model in which auxin inhibits CUC1 expression, but CUC1 promotes PIN1 polarization to promote auxin maxima formation. The same module has been used to explain leaf margin patterning, and the authors show that this module explains patterning robustness. They showed that this module promotes faster growth in primordia, and partially breaking the module, such as reducing CUC1 expression, slows down primordia growth, leading to robust patterning.

Much is known about genes and proteins as parts of life, as well as their biochemical functions. However, we lack a coherent understanding of how they assemble and operate. This study shed new light on the cybernetics underlying patterning in plants. The paper is in general well-written, and the images and movies are of high quality. My have only a few questions.

1. How CUC1 promotes PIN1 polarization? Is it through promoting PIN1 gene expression?
2. Figure 4a, DR5 is not seen in the center of floral meristems. However, DR5 should be there at an early stage. How about CUC1 expression by that stage?

Reviewer #3 (Remarks to the Author):

This paper investigates how robustness is controlled during organ development. The authors show that in the *drmy1* mutant of *Arabidopsis*, CUC1 expression and auxin response are both observed more broadly, and that suppression of auxin synthesis and transport also cause diffuse CUC1 expression, suggesting that disrupted auxin maxima cause altered CUC1 localisation. They show that plants where the CUC regulator miRNA site has been mutated produce variable petal numbers and initiation like the *drmy1* mutant, and that the *drmy1cuc1* double mutant produces much more robust sepal numbers, suggesting that CUC1 underlies variable petal number in *drmy1*. They also show that overexpression of CUC1 increases auxin maxima intensity, and that CUC1 overexpression causes additional auxin maxima to form, potentially underlying the loss of robustness in sepal number. Finally, they show that much of this behaviour can be explained by a computational model where CUC1 increases PIN1 polarisation.

Overall, the authors have nicely combined biological experiments and computational modelling to generate a plausible hypothesis for how sepal number robustness is maintained. The findings are interesting with broad implications for understanding developmental robustness, and the data largely support the conclusions. I have some suggestions to improve the manuscript below, most importantly additional tests of some of the interesting model predictions:

1- The computational modelling makes several interesting predictions. Many are consistent with the genetic and developmental data presented, and serve to test hypotheses generated there. However, there are many aspects of the model that are speculative and some of these should be tested biologically to confirm the model's overall accuracy. At the moment it risks appearing bolted onto the end of a paper without having its considerable insights tested.

For example:

The model predicts that cell division rate will alter noise in auxin levels. This could be tested biologically by observing noise in auxin levels in different tissues that are growing at different rates, or in mutants with different growth and division rates. It should also be possible to rescue the variable *drmy1* mutant by crossing to mutants with lower overall growth rates.

The model also predicts that CUC1 enhances the repolarisation of PIN substantially. This could potentially be tested by performing a perturbation (eg. Cell ablation or ectopic auxin application) and observing the speed of PIN reorientation in tissues with different CUC levels, perhaps in wild type, *cuc1* mutants and CUC overexpressors. More generally, does PIN repolarisation in vivo fit with model predictions? The authors do not currently image it.

2- The idea that CUC1 increases the sensitivity of PIN to fluctuating auxin levels is interesting. What molecular/ mechanical mechanism do the authors propose for this?

3- The "speed vs. robustness" trade off that the authors propose may well be correct, and is very interesting if it is. However, it is currently supported only by modelling and not by experimental data. It would greatly improve the paper to analyse sepal number in plants with increased or decreased growth rate to understand if this affects sepal number robustness.

4- Previous work has suggested that CUC genes repress growth in boundary domains to aid morphogenesis. How do the results here fit with this idea? Given that the data here suggest that reducing growth rate should increase robustness, but overexpression of CUC actually decreases robustness it seems particularly important to understand how to integrate these hypotheses. Does CUC1 have an effect on growth rate in the context of the sepal, or just on the reorientation of PIN? The data in fig. 3 seem to show that *cuc* mutants have smaller, and 5mCUC plants have larger, buds. Understanding the implications of these different roles of CUC in an integrated fashion would make this paper particularly important for the field.

5- Is this mechanism for robustness specific to sepals, or does it act in other organs too eg. Other floral organ whorls, or leaves produced at the SAM? i.e. is *drmy* more variable in other organ numbers and angles too?

6- Diagram j in fig. 6 is unclear. The authors propose a trade off between robustness and speed driven by CUC levels, but it appears that this tradeoff is only present in a background of high noise, based on the figure. The data from the high CUC activity 5mCUC lines are also missing, which would lend more weight to the speed/ robustness argument in this diagram. Otherwise the authors should make clear in the results and discussion that the speed/ robustness trade off is only present in a background of high developmental noise.

**Response to Reviewers' Comments on Nature Communications Manuscript NCOMMS-23-58512
"Tradeoff Between Speed and Robustness in Primordium Initiation Mediated by Auxin-CUC1
Interaction"
May 1, 2024**

We would like to thank the reviewers for their thoughtful suggestions and comments. We have done additional experiments and revised the manuscript accordingly. We have addressed each comment as described below.

Reviewer #1 (Remarks to the Author):

The manuscript entitled "Tradeoff Between Speed and Robustness in Primordium Initiation Mediated by Auxin-CUC1 Interaction" by Kong et al. describes the role of DRMY1-CUC1-Auxin in control of speed/robustness of floral organ formation by a precise manner. The authors show that CUC1 is upregulated in the *drmy1* background. Removing of CUC1 may rescue the noise problem of the auxin signaling, while this may lead to a decreased speed of sepal initiation. The idea of the balance between speed and robustness in organ initiation seems to be very interesting. Overall, the data are very clearly presented and the manuscript is well organized.

We thank Reviewer #1 for the accurate summary and appreciation of our work.

Specific points:

(1) The authors also tested the CUC2 expression pattern (Fig S1) and showed a similar result as CUC1. However, CUC2 could not function as CUC1 in genetic experiments. How about CUC3? Since the three CUC genes may play redundant roles in organ initiation, what is the unique molecular function for CUC1? Can CUC2 replace CUC1 in the *cuc1* mutant background? The authors may have a deeper discussion on the CUC genes.

Response: We think that CUC1 plays a more important role than CUC2 and CUC3 during sepal initiation and contributes more to the loss of robust sepal initiation in the *drmy1* mutant for the following reasons:

- In mid-stage 2 floral meristem, both *CUC1* and *CUC2* are strongly expressed in the boundaries between incipient sepal primordia and the center of floral meristem (Fig. 1d; Supplementary Fig. 1a). However, at slightly earlier stages when *CUC1* expression is just starting to concentrate at these boundaries (see next page, arrowheads), *CUC2* expression is weak and diffuse in the floral meristem. At that stage, *CUC2* is strongly expressed in the boundary between the floral meristem and the inflorescence meristem (see next page, arrows), where *CUC1* is lowly expressed. Similar to *CUC2*, in previously published micrographs, *CUC3* is strongly expressed in the boundary between the floral meristem and the inflorescence meristem, as well as abaxial-most regions of the bud, and barely in the floral meristem-sepal boundary¹ (in this reference see Fig. S4C). Therefore it is likely that *CUC1* mainly functions in specifying the floral meristem-sepal boundary (which *CUC2* expression follows), and *CUC2* and *CUC3* mainly functions in specifying the inflorescence meristem-floral meristem boundary (where *CUC1* expression is low).

- In our RNA-seq using the induced *ap1 cal AP1-GR* inflorescence, we found an upregulation of *CUC1*, but not *CUC2* and *CUC3*, in the *drmy1* mutant (Fig. 1c). Mutation of *cuc1* restores robustness in sepal primordium number and position (Fig. 2f-k), while mutation of *cuc2* does not (Supplementary Fig. 3i-n). These results support the idea that *CUC1* is more important than *CUC2* in the specification of sepal boundaries, and that the upregulation and expanded expression domain of *CUC1*, but not of *CUC2*, is responsible for the loss of developmental robustness in *drmy1* mutant sepals.
- In addition, the *cuc1* single mutant has reduced auxin maxima intensity (Fig. 4a-b) and reduced speed of sepal initiation (Fig. 3d-f). These results show that *CUC1* has its unique function during sepal initiation, and that *CUC2* cannot replace *CUC1* in the *cuc1* mutant background.

Despite these reasons, we cannot fully exclude the possibility that *CUC2* and *CUC3* play a small role in sepal initiation alongside *CUC1*. Thus, we have summarized these points and added the following sentences to our Discussion section: “While we showed that the *cuc2* mutation did not restore developmental robustness to *drmy1* (Supplemental Fig. 3i-n), and previously published micrographs showed that *CUC3* was expressed at a very low level in the floral meristem-incipient sepal boundary¹, we cannot exclude the possibility that *CUC2* and *CUC3* are also involved in this tradeoff alongside *CUC1*.”

(2) How about the upregulation of *CUC2/CUC3*? Could the upregulation of *CUC2/CUC3* in regulation of organ initiation be similar to *CUC1* upregulation?

Response: In Ler background, simultaneous upregulation of *CUC1*, *CUC2*, and *CUC3* (along other miR164 targets) in the *mir164abc* mutant results in stronger variability in sepal initiation than upregulation of *CUC1* alone (*CUC1m-GFP*). It is therefore likely that upregulation of *CUC2/CUC3* is synergistic to the upregulation of *CUC1* in disrupting robustness in sepal initiation.

(3) I found that some of the data in Supplemental Figures are important for the story, for examples some data in Figure S6 and S7. Is it possible for some of them to be moved to the main text?

Response: Thank you for your suggestion. We have moved the original Figure S6 to the beginning of Figure 7 (panels a-c, f), to which we appended experimental validation of the model prediction (panels d, e, g-i).

Reviewer #2 (Remarks to the Author):

The manuscript by Kong et al. entitled “Tradeoff Between Speed and Robustness in Primordium Initiation Mediated by Auxin-CUC1 Interaction” reports a role of the auxin-CUC module in patterning robustness. In the *drmy1* mutant, in which sepal formation is less synchronized and can have variable number of sepals, the authors found that CUC1 expression was increased, and reducing CUC1 expression alleviates the phenotype. They also found that overexpressing CUC1 similarly reduced the robustness of sepal formation. Through live imaging, the authors showed that CUC1 expression promotes sepal primordium growth rate in a non-cell-autonomous manner, although CUC1 per se reduces cell growth. Auxin signaling maxima were also found enhanced, and ectopic auxin signaling, which forms proceed of sepal primordium formation, was observed in CUC1 overexpression lines and *drmy1*. The authors proposed a model in which auxin inhibits CUC1 expression, but CUC1 promotes PIN1 polarization to promote auxin maxima formation. The same module has been used to explain leaf margin patterning, and the authors show that this module explains patterning robustness. They showed that this module promotes faster growth in primordia, and partially breaking the module, such as reducing CUC1 expression, slows down primordia growth, leading to robust patterning.

Much is known about genes and proteins as parts of life, as well as their biochemical functions. However, we lack a coherent understanding of how they assemble and operate. This study shed new light on the cybernetics underlying patterning in plants. The paper is in general well-written, and the images and movies are of high quality. My have only a few questions.

We thank Reviewer #2 for these nice comments on our manuscript.

1. How CUC1 promotes PIN1 polarization? Is it through promoting PIN1 gene expression?

Response: Following your suggestion and a similar suggestion by Reviewer #3, we have imaged the PIN1-GFP reporter in WT vs. *cuc1* mutant (Supplementary Fig. 6). We found that the PIN1 protein accumulates at a similar level in WT and the *cuc1* mutant, and their polarity pattern is similar in late stage 2 (Supplementary Fig. 6c-d). However, PIN1 is polarized towards the incipient sepal primordia at an earlier stage in WT than *cuc1*, suggesting that CUC1 promotes PIN1 polarization (Supplementary Fig. 6a-b). We further performed ablation and observed that PIN1 reorients away from the ablation site faster in WT than *cuc1* (Supplementary Fig. 6e-h). These results suggest that CUC1 does not increase the expression of *PIN1* but instead makes it more dynamic and more sensitive in response to reorientation cues.

2. Figure 4a, DR5 is not seen in the center of floral meristems. However, DR5 should be there at an early stage. How about CUC1 expression by that stage?

Response: In stage 1 when the floral meristem has just initiated from the inflorescence meristem, DR5 is seen in the center of the floral meristem. At that stage, *CUC1* is expressed at the two sides where the floral meristem joins the inflorescence meristem, as revealed by both the transcriptional reporter and protein reporter of CUC1 (see images below; dotted lines mark the outline of the stage 1 meristem).

Reviewer #3 (Remarks to the Author):

This paper investigates how robustness is controlled during organ development. The authors show that in the *drmy1* mutant of arabidopsis, CUC1 expression and auxin response are both observed more broadly, and that suppression of auxin synthesis and transport also cause diffuse CUC1 expression, suggesting that disrupted auxin maxima cause altered CUC1 localisation. They show that plants where the CUC regulator miRNA site has been mutated produce variable petal numbers and initiation like the *drmy1* mutant, and that the *drmy1 cuc1* double mutant produces much more robust sepal numbers, suggesting that CUC1 underlies variable petal number in *drmy1*. They also show that overexpression of CUC1 increases auxin maxima intensity, and that CUC1 overexpression causes additional auxin maxima to form, potentially underlying the loss of robustness in sepal number. Finally, they show that much of this behaviour can be explained by a computational model where CUC1 increases PIN1 polarisation.

Overall, the authors have nicely combined biological experiments and computational modelling to generate a plausible hypothesis for how sepal number robustness is maintained. The findings are interesting with broad implications for understanding developmental robustness, and the data largely support the conclusions.

We thank Reviewer #3 for the precise summary and nice comments of our manuscript.

I have some suggestions to improve the manuscript below, most importantly additional tests of some of the interesting model predictions:

1- The computational modelling makes several interesting predictions. Many are consistent with the genetic and developmental data presented, and serve to test hypotheses generated there. However, there are many aspects of the model that are speculative and some of these should be tested biologically to confirm the model's overall accuracy. At the moment it risks appearing bolted onto the end of a paper without having its considerable insights tested.

For example:

The model predicts that cell division rate will alter noise in auxin levels. This could be tested biologically by observing noise in auxin levels in different tissues that are growing at different rates, or in mutants with different growth and division rates. It should also be possible to rescue the variable *drmy1* mutant by crossing to mutants with lower overall growth rates.

Response: We thank you for making this suggestion.

To test the prediction that decreasing tissue growth rate restores developmental robustness, we cultured WT and *drmy1* mutant buds on low sucrose (0.1% w/v) media, compared to normal culture condition with 1% (w/v) sucrose. The low sucrose media decreases bud growth rate by half (Supplementary Fig. 7a). In contrast to noise amplification in *drmy1* buds under normal culture condition, *drmy1* buds cultured on low sucrose media show noise dampening and robust DR5 pattern (Fig. 7e, h). Robustness in sepal initiation pattern is also restored (Fig. 7e, i, Supplementary Fig. 7b).

These experiments confirmed our model prediction that decreasing tissue growth rate restores developmental robustness in the *drmy1* mutant. They highlight the tradeoff between morphogenesis speed and robustness which is the gist of our manuscript.

The model also predicts that CUC1 enhances the repolarisation of PIN substantially. This could potentially be tested by performing a perturbation (eg. Cell ablation or ectopic auxin application) and observing the speed of PIN reorientation in tissues with different CUC levels, perhaps in wild type, *cuc1* mutants and CUC overexpressors. More generally, does PIN repolarisation *in vivo* fit with model predictions? The authors do not currently image it.

Response: Thank you for suggesting these experiments.

Following your suggestion, we have crossed the PIN1-GFP reporter with the *cuc1* mutant, and imaged PIN1-GFP polarity patterns *in vivo* in WT vs. *cuc1* mutant (Supplementary Fig. 6a-d). We found that, in mid-stage 2, PIN1-GFP is strongly polarized towards incipient sepal primordia in WT, while this polarity pattern is less clear in *cuc1*. In late stage 2, both WT and *cuc1* mutant buds show strong polarity pattern towards the incipient sepal primordia. These results indicate that CUC1 does not affect the final polarity pattern but increases its polarization speed.

To further test this idea, we have performed ablation in WT vs. *cuc1* mutant floral meristems carrying the PIN1-GFP reporter, and live imaged them every hour. We found that PIN1-GFP reorients faster in WT than *cuc1* mutant in response to the ablation. Specifically, 60% of the cells completed reorientation 1 hour after ablation in WT, while most cells complete reorientation within 2-3 hours in *cuc1* mutant (Supplementary Fig. 6e-h). These results agree with our hypothesis that CUC1 increases the reorientation speed of PIN1.

Finally, we attempted lanolin paste application of 10 mM IAA on the floral meristem (see images below; these are WT meristems carrying the PIN1-GFP reporter; asterisks indicate the IAA application site). We pre-treated buds with L-Kynurenine for 2 days to deplete endogenous auxin (top row), or L-Kynurenine and Naphthylphthalamic acid for 2 days to deplete endogenous auxin and inhibit polar auxin transport so that the applied IAA stays at the application site (bottom row). In either case, we did not see a convincing PIN1 reorientation within 6 hours of IAA application even in WT. We therefore did not use this experiment to compare PIN1 reorientation speed in WT vs. *cuc1* mutant.

Overall, our *in vivo* observation of PIN1-GFP and the ablation experiment support our idea that CUC1 makes PIN1 more dynamic and more responsive to reorientation cues.

2- The idea that CUC1 increases the sensitivity of PIN to fluctuating auxin levels is interesting. What molecular/ mechanical mechanism do the authors propose for this?

Response: Thank you for your suggestion. We have discussed possibilities of how CUC1 affects PIN repolarization in our Discussion section: “Our computational modeling suggest that the speed-robustness tradeoff can be fully explained by the function of CUCs in increasing PIN1 repolarization, which was previously reported in other developmental contexts^{25,39} and tested here in the floral meristem (Supplementary Fig. 6). How CUCs increase PIN1 repolarization remains unknown. It was shown that polarity of PIN proteins can be regulated by phosphorylation (e.g., PID⁴⁸, D6PK⁴⁹, and PP2A⁵⁰) or membrane trafficking (e.g., ABCB19⁵¹ and ROP2⁵²). Thus, CUCs may increase PIN1 repolarization by changing the expression of these important PIN regulators. Alternatively, CUCs may also inhibit growth, causing mechanical conflict with adjacent fast-growing regions which alters PIN1 polarity⁴⁰. Further study is needed to test whether CUCs increase PIN1 repolarization by any of these mechanisms.”

3- The “speed vs. robustness” trade off that the authors propose may well be correct, and is very interesting if it is. However, it is currently supported only by modelling and not by experimental data. It would greatly improve the paper to analyse sepal number in plants with increased or decreased growth rate to understand if this affects sepal number robustness.

Response: As mentioned above, we followed your suggestion and cultured *drmy1* mutant buds under low sucrose media, which slows down tissue growth. We found that reduced tissue growth rate restores robustness in sepal number and positioning in *drmy1* (Fig. 7e, i, Supplementary Fig. 7b).

4- Previous work has suggested that CUC genes repress growth in boundary domains to aid morphogenesis. How do the results here fit with this idea? Given that the data here suggest that reducing growth rate should increase robustness, but overexpression of CUC actually decreases robustness it seems particularly important to understand how to integrate these hypotheses. Does CUC1 have an effect on growth rate in the context of the sepal, or just on the reorientation of PIN? The data in fig. 3 seem to show that *cuc* mutants have smaller, and *5mCUC* plants have larger, buds. Understanding the implications of these different roles of CUC in an integrated fashion would make this paper particularly important for the field.

Response: In the model (Fig. 7b-c) and the newly added low sucrose treatment (Fig. 7d-e), developmental robustness is restored when *global* growth rate is reduced. From our live imaging, CUC1 does not seem to have an effect on global growth rate (Fig. 3). Comparing WT vs. *5mCUC1*, and WT vs. *cuc1*, bud diameters increase by approximately the same amount from the start till the end of the live imaging.

In the context of organ initiation, CUC genes are highly expressed in boundary regions where they repress growth. They also increase PIN sensitivity and polar auxin transport, making stronger auxin maxima. Faster growth at these auxin maxima, combined with growth suppression at the organ boundary, means faster organ initiation in the presence of CUC genes.

Thus, we think that the spatiotemporal manner of growth suppression, and the precise relative localization of fast-growth and slow-growth regions, are important to make the distinction between the two scenarios. While global growth inhibition improves developmental robustness, boundary expression of *CUC1* can amplify auxin noise. Amplified auxin noise triggers growth, and expression of CUC1 promotes formation of boundaries surrounding the auxin noise, which together give rise to variably initiated sepals.

5- Is this mechanism for robustness specific to sepals, or does it act in other organs too eg. Other floral organ whorls, or leaves produced at the SAM? i.e. is *drmy1* more variable in other organ numbers and angles too?

Response: The *drmy1* mutation also affect the robustness in the initiation of floral meristem and other floral organs besides sepals. In our previous publication², we have shown that the *drmy1* mutant has increased variability in the number of petals and stamens (see Extended Data Fig. 1a in that reference), in addition to sepals. We have also shown that phyllotaxy pattern is disrupted in the *drmy1* mutant (see Extended Data Fig. 6j in that reference).

6- Diagram j in fig. 6 is unclear. The authors propose a trade off between robustness and speed driven by CUC levels, but it appears that this tradeoff is only present in a background of high noise, based on the figure. The data from the high CUC activity *5mCUC* lines are also missing, which would lend more weight to the speed/ robustness argument in this diagram. Otherwise the authors should make clear in the results and discussion that the speed/ robustness trade off is only present in a background of high developmental noise.

Response: Following your suggestion, we have added data from *5mCUC1* lines to the final diagram (now Fig. 8). In the new model, higher *CUC1* expression increases the speed of sepal initiation but disrupts its robustness. In *5mCUC1* which has the highest *CUC1* expression, sepal initiation is variable even under low auxin noise (Fig. 4a, Supplementary Fig. 5). This variability can be further increased when *5mCUC1* is in the high auxin noise condition (i.e. treated with BAP, which provides much more initial auxin noise; Fig. 5f).

References

1. Fal, K. *et al.* Phyllotactic regularity requires the *paf1* complex in Arabidopsis. *Dev.* **144**, 4428–4436 (2017).
2. Zhu, M. *et al.* Robust organ size requires robust timing of initiation orchestrated by focused auxin and cytokinin signalling. *Nat. Plants* **6**, 686–698 (2020).

REVIEWERS' COMMENTS

Reviewer #1 (Remarks to the Author):

The authors have addressed most of my concerns. I have no further suggestions.

Reviewer #2 (Remarks to the Author):

The authors have well addressed comments from myself and the other reviewers. I have no further comments except a reminder that CUC3 is not under miR164 regulation.

Reviewer #3 (Remarks to the Author):

The authors have addressed all of my concerns in full. The additional experiments beautifully test and confirm the interesting predictions of their model and the paper will be an outstanding addition to the field.